# TORCHRL: A DATA-DRIVEN DECISION-MAKING LIBRARY FOR PYTORCH

**Albert Bou**
UPF, Acellera

**Matteo Bettini**
University of Cambridge

**Sebastian Dittert**
UPF

**Vikash Kumar**
Meta AI

**Shagun Sodhani**
Meta AI

**Xiaomeng Yang**
Meta AI

**Gianni De Fabritiis**
ICREA, UPF, Acellera

**Vincent Moens**
PyTorch, Meta
`vmoens[at]meta.com`

## ABSTRACT

PyTorch has ascended as a premier machine learning framework, yet it lacks a native and comprehensive library for decision and control tasks suitable for large development teams dealing with complex real-world data and environments. To address this issue, we propose TorchRL, a generalistic control library for PyTorch that provides well-integrated, yet standalone components. We introduce a new and flexible PyTorch primitive, the TensorDict, which facilitates streamlined algorithm development across the many branches of Reinforcement Learning (RL) and control. We provide a detailed description of the building blocks and an extensive overview of the library across domains and tasks. Finally, we experimentally demonstrate its reliability and flexibility and show comparative benchmarks to demonstrate its computational efficiency. TorchRL fosters long-term support and is publicly available on GitHub for greater reproducibility and collaboration within the research community. The code is open-sourced on GitHub.

## 1 INTRODUCTION

Originally, breakthroughs in AI were powered by custom code tailored for specific machines and backends. However, with the widespread adoption of AI accelerators like GPUs and TPUs, as well as user-friendly development frameworks such as PyTorch (Paszke et al., 2019), TensorFlow (Abadi et al., 2015), and Jax (Bradbury et al., 2018), research and application has moved beyond this point. In this regard, the field of decision-making is more fragmented than other AI domains, such as computer vision or natural language processing, where a few libraries have rapidly gained widespread recognition within their respective research communities. We attribute the slower progress toward standardization to the dynamic requirements of decision-making algorithms, which create a trade-off between modularity and component integration.

The current solutions in this field lack the capability to effectively support its wide range of applications, which include gaming (Mnih et al., 2013; Silver et al., 2016; OpenAI et al., 2019b; Espeholt et al., 2018) , robotic control (Tobin et al., 2017; OpenAI et al., 2019a), autonomous driving (Aradi, 2022) , finance (Ganesh et al., 2019) , bidding in online advertisement (Zhu & Roy, 2021), cooling system control (Luo et al., 2022), faster matrix multiplication algorithms (Fawzi et al., 2022) or chip design (Mirhoseini et al., 2021). Decision-making is also a powerful tool to help train other models through AutoML solutions (He et al., 2018; Zoph & Le, 2017; Zoph et al., 2018; Baker et al., 2017; Zhong et al., 2018) or incorporate human feedback in generative model fine-tuning (Christiano et al., 2017; von Werra et al., 2023; Nakano et al., 2021), and is also used in techniques such as sim-to-real transfer (Tobin et al., 2017; Peng et al., 2018; Rudin et al., 2022; OpenAI et al., 2019a), model-based RL (Brunnbauer et al., 2022; Wu et al., 2022; Hansen et al., 2022a;b) or offline data collection for batched RL and imitation learning (Kalashnikov et al., 2018; Shah et al., 2022). In addition, decision-making algorithms and training pipelines must often meet challenging scaling requirements, such as distributed inference (Mnih et al., 2016; Espeholt et al., 2018) or distributed model execution (Nakano et al., 2021). They must also satisfy imperative deployment demands when used on hardware like robots or autonomous cars, which often have limited computational

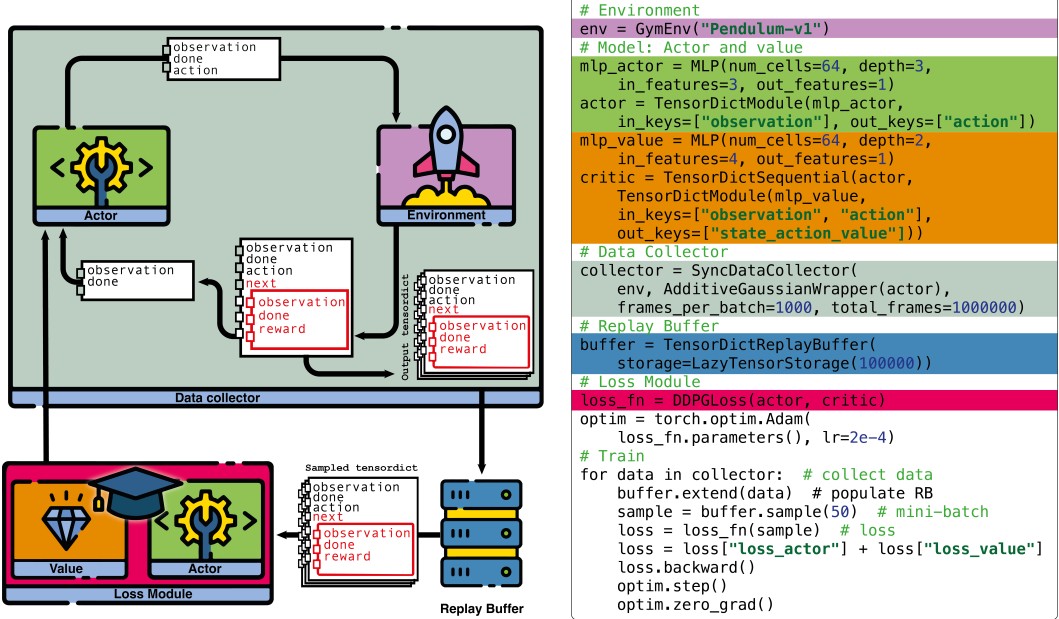

Figure 1: TorchRL overview. The left side showcases the key components of the library, demonstrating the data flow with TensorDict instances passing between modules. On the right side, a code snippet is provided as a toy example, illustrating the training of DDPG. The script provides users with full control over the algorithm's hyperparameters, offering a concise yet comprehensive solution. Still, replacing a minimal number of components in the script enables a seamless transition to another similar algorithm, like SAC or REDQ.

resources and need to operate in real-time. Many libraries are either too high-level and difficult to repurpose across use cases, or too low-level for non-experts to use practically. Because of this intricate landscape, a consensus adoption of frameworks within the research community remains limited (Brockman et al., 2016). We believe that moving forward, the next generation of libraries for decision-making will need to be easy to use and rely on widespread frameworks while still being powerful, efficient, scalable, and broadly applicable, to meet the demands of this rapidly evolving and heterogeneous field. This is the problem that TorchRL addresses.

Training decision-making models involves a sequentially driven algorithmic process, where the cooperation between framework components is vital for adaptability across different applications. For example, actors, collectors and replay buffers need to be able to handle various modalities. Imposing a fixed input/output format would restrict their reusability while eliminating all restrictions would create significant difficulties in designing adaptable scripts. Consequently, conventional decision-making libraries must choose between a well-integrated framework with a high-level API or a flexible framework with basic building blocks of limited scope. The former hinders quick coding and testing of new ideas, while the latter leads to complex code and a poor user experience. TorchRL effectively tackles these challenges with a powerful communication class that allows its components to operate independently while not limiting their scope and at the same time ensuring seamless integration with each other to flexibly create algorithms.

In this paper, we will detail our approach for a truly general decision-making library, based primarily on module portability, efficient communication and code repurposing. We introduce TensorDict, a new PyTorch data carrier, which is central to achieving this goal. By adopting this new coding paradigm, we demonstrate how simple it is to reuse ML scripts and primitives across different tasks or research domains. Building upon this innovative technical foundation, we explore various independent components that rely on this abstraction. These components encompass a range of advanced solutions, including distributed data collectors, cutting-edge models like Decision Transformers, and highly specialized classes such as an environment API compatible with single and multi-agent problems. We demonstrate that the current state of the library encompasses diverse categories of decision-making algorithms, caters to a wide range of applications, performs well on both small-scale machines and

distributed clusters, and strikes a balance in terms of technical requirements that makes it accessible to practitioners with multiple levels of expertise.

**Related work.** Currently, there is a vast array of libraries available for reinforcement learning. Some of these libraries are specifically designed for particular niches, such as Pyqlearning (Accel-Brain, 2023) and Facebook/ELF (Tian et al., 2017), only support a small number of environments, like OpenSpiel(Lanctot et al., 2019) and reaver (Ring, 2018), or lack the capability for parallel environments with distributed learning, such as MushroomRL (D'Eramo et al., 2021). Additionally, many previously popular libraries are no longer actively maintained, including, KerasRL (Plappert, 2016), rlpyt (Stooke & Abbeel, 2019), ReAgent (Gauci et al., 2018), IntelLabs/coach (Caspi et al., 2017), Dopamine (Castro et al., 2018), ChainerRL (Fujita et al., 2021), PyTorchRL (Bou et al., 2022), TFAgents (Guadarrama et al., 2018), and autonomous-learning-library(Nota, 2020). Therefore, for libraries aspiring to serve as comprehensive tools for decision-making, it is crucial to offer robust long-term support while also being designed to ensure adaptability to the continuously evolving landscape of decision-making research.

Among currently popular solutions, there's a strong trend toward high-level approaches that make it easier to integrate different parts. This is particularly necessary for tools that have components that can't stand alone or rely on limited data carriers and require some extra code to make everything work smoothly together. Some well-known examples of these tools include Stable-baselines (an Stable-baselines3, SB3) (Raffin et al., 2021), ElegantRL (Liu et al., 2021), Acme (Hoffman et al., 2020), DI-engine (engine Contributors, 2021), garage (garage contributors, 2019), Tensorforce (Kuhnle et al., 2017), and RLlib (Liang et al., 2018). While high-level approaches can be welcoming to newcomers, they often obscure the inner workings of the code, which may not suit users who seek greater control over their scripts, and difficult code repurposing. TorchRL leverages its powerful data carrier to adopt a highly modular design. Our primary goal is to provide well-tested building blocks that enable practitioners to create adaptable scripts. These building blocks can also be utilized by high-level tools to cater to users of all experience levels, avoiding the need to reimplement the same functionalities again. Finally, two additional popular libraries are Tianshou (Weng et al., 2022) and CleanRL (Huang et al., 2022). Tianshou shares similarities with TorchRL, offering a wide range of RL algorithms, a relatively versatile data carrier, and modular components. However, its primary focus is on building RL trainers to simplify problem-solving without extensive domain knowledge, which we believe can be complementary to our approach. On the other hand, CleanRL provides efficient single-file implementations of RL algorithms in Torch and JAX, enhancing code clarity. Yet, it sacrifices component reusability, making scaling challenging due to code duplication.

## 2 TORCHRL COMPONENTS

TorchRL is made up of separate, self-contained parts. Each of these parts, referred to as components, handles a specific operation in the overall data workflow. While it's not always required, these components usually communicate using a dedicated tool designed to flexibly pass data back and forth, the TensorDict. The resulting atomicity and modularity of this approach facilitate versatile usage and a combination of individual components within most machine learning workflows. In this section, we provide an overview of the key components in the library, which can be used independently or as foundational elements for constructing decision-making algorithms, as depicted in Figure 1.

**TensorDict**[1]**.** Introducing a seamless mode of communication among independent RL algorithmic components poses a set of challenges, particularly when considering the diversity of method signatures, inputs and outputs. A shared, versatile communication standard is needed in such scenarios to ensure fluid interactions between modules without imposing constraints on the development of individual components. We address this problem by adopting a new data carrier for PyTorch named TensorDict, packaged as a separate open-source lightweight library. This library enables every component to be developed independently of the requirements of the classes it potentially communicates with. As a result, data collectors can be designed without any knowledge of the policy structure, and replay buffers do not need any information about the data structure that is to be stored within them. Figure 1 gives an overview of a practical data flow using TensorDict as a communication tool.

---

[1]https://github.com/pytorch/tensordict

A TensorDict is a dictionary-like object that stores tensors(-like) objects. It includes additional features that optimize its use with PyTorch. One of its notable features is the ability to handle the batch size (in contrast with the "feature" size), which means that it can be indexed not only by keys but also by shapes. The adopted convention is that a TensorDict batch size represents the common leading dimensions of the tensors it contains. In the context of RL, the batch-size can include the minibatch or time dimension as well as the number of agents, tasks or processes. The shape-key duality is reflected in TensorDict's ability to handle both key-based and numerical indexing. For both, this indexing can be performed along multiple dimensions (through nested tensordicts along the key-dimensions, and through the multiple tensor dimensions along the shape-dimension).

TensorDict provides a whole stack of extra functionality that naturally follows from these two basic features. These can roughly be split into shape-based and key-based methods. In the first bucket, one can find `torch.Tensor`-like features such as reshaping, (un)sequeezing, stacking, concatenating, masking or permuting. The latter contains methods that allow to flatten or unflatten the tensordict structure (to represent a flat structure as a nested one or inversely), renaming or deleting keys. Other utilities include methods to move a TensorDict from one device to another, point-to-point communication in distributed settings and many more. The class also offers a set of efficient storage interfaces through memory-mapped tensors or HDF5 files, allowing to manipulate big data structures with little effort. More details about TensorDict functionality, as well as benchmarks against comparable solutions, can be found in Appendix B.

TensorDict enables the design of functions and classes with a generic signature, where we can safely restrict the input and output to be both TensorDict instances. This comes in handy in countless cases: a straightforward example is the support that TorchRL provides for different simulator backends such as Gym (Brockman et al., 2016), IsaacGym (Makoviychuk et al., 2021), DeepMind Control (Tunyasuvunakool et al., 2020), Brax (Freeman et al., 2021) and others. Each of these libraries has a dedicated `step` signature that is unified under the TorchRL `EnvBase` where the relative method expects a TensorDict instance as input and writes the data in it in a standardized format. The input types supported are almost universal, which allows TorchRL's environments to receive complex data structures as input, thereby enabling a common API for stateful and stateless environment. Regarding the outputs, the `info` dictionary returned by Gym-like environment will is expanded in the output, making the whole `step` results readily available in a consistent way across libraries and environments. In practice, this means that in TorchRL, the same training script can be repurposed with minimal effort to train environments from various backends and domains.

TensorDict does not only provide clearer code but also comes with some performance improvements that include better memory management, more efficient (a)synchronous point-to-point communication in distributed settings and device casting. Finally, the `tensordict` library also comes with a dedicated `tensordict.nn` package that rethinks the way one designs PyTorch models. Various primitives, such as `TensorDictModule` and `TensorDictSequential` allow to design of complex PyTorch operations in an explicit and programmable way. TensorDict's `nn` primitives are fully compatible with `torch.compile` through a dedicated symbolic tracer available in the TensorDict package, making TorchRL itself compatible with the latest features of PyTorch 2.1.

**Environment API, wrappers, and transforms.** OpenAI Gym (Brockman et al., 2016) has become the most widely adopted environment interface in RL: its standardization of the environment API was a major leap forward that enabled research reproducibility. Its simplicity (only two functions are exposed during interaction) drove its success, but it presents several drawbacks, such as a fixed tuple signature for its `step` method or its heavy reliance on wrappers (see below).

The TorchRL environment interface aims to maintain the simplicity of Gym while addressing these drawbacks. As with Gym, only the `reset()` and `step()` methods are required to interact with these objects. Both of these methods rely on TensorDict to facilitate their integration, as discussed in the TensorDict section and presented Figure 1. TensorDicts enable carrying multidimensional tensor-like data, allowing batched/vectorized simulation (Freeman et al., 2021; Bettini et al., 2022) and multidimensional input/output spaces. The native support for batched data shapes and nested data structures make TorchRL compatible by default with multi-agent and multi-task applications, where these features are key for a clear environment API.

Consequently, TorchRL is by no means an extension of Gym or any other simulation library, unlike other libraries that cover one specific simulator but not others. The generic environment API allows to easily support a multitude of existing simulators: Gym and Gymnasium (Brockman et al., 2016) since

v0.13, DMControl (Tunyasuvunakool et al., 2020), Habitat (Szot et al., 2021), RoboHive (Rob, 2020), OpenML datasets (Vanschoren et al., 2013), D4RL datasets (Fu et al., 2020), Brax (Freeman et al., 2021), Isaac (Makoviychuk et al., 2021), Jumanji (Bonnet et al.), and VMAS (Bettini et al., 2022) are some examples, but the list is constantly growing. Notably, the last four of these are vectorized and can pass gradients through the simulations, allowing to compute reparameterized trajectories. We provide some more technical information about the environment API and its usage in Appendix C.

TorchRL draws inspiration from other components of the PyTorch ecosystem (maintainers & contributors, 2016; Paszke et al., 2019) which rely on the concept of transform sequences to modify module outputs. Significantly, TorchRL's Transforms, which should be seen as regular `nn.Module` instances that can be applied wherever data transformation is required. Transforms are versatile and can be utilized in various components, including replay buffers, collectors, and even ported from environment to models, effectively connecting the training process with real-world applications. Examples include data transformations (e.g. resizing, cropping), target computation (reward-to-go) or even embedding through foundational models (Nair et al., 2022; Ma et al., 2022). As we show in the Appendix, transforms offer a way to dynamically manipulate data at least as flexible as wrapping classes typically used in RL libraries.

**Data collectors.** TorchRL has dedicated classes for data collection that execute a policy in one or multiple batched environments and return batches of transition data in TensorDict format. Data collectors iteratively compute the actions to be executed, pass those to the environments (real or simulated), and can handle resetting when and where required. These collectors are designed for ease of use, requiring only one or more environment constructors, a policy module and a target number of steps. However, developers can also specify the asynchronous or synchronous nature of data collection, the number of parallel environments, the resource allocation (e.g., GPU, number of workers) and the data postprocessing, giving them fine control and flexibility over the collection process. TorchRL offers distributed components that enable data collection at scale by coordinating multiple workers in a cluster. These components are compatible with various backends like gloo or NCCL through torch.distributed, and multiple launchers and resource management solutions including submitit (Incubator, 2021) and Ray (Moritz et al., 2018). As scaling inevitably adds complexity, the distributed solutions in TorchRL are designed as independent components that provide the same interface and data control as non-distributed components. This enables practitioners to work locally on projects with complete independence while also providing an easy way to scale up to distributed projects by just replacing non-distributed components with distributed counterparts, which will increase data collection throughput. Crucially, TorchRL distributed dependencies are optional and not required for non-distributed components.

**Replay buffers and datasets.** Replay buffers (RBs) are crucial elements of RL that enable agents to learn from past experiences by storing, processing, and resampling data. However, creating a flexible RB class that caters to various use cases without duplicating implementations can be challenging. To overcome this, TorchRL provides a single RB implementation that offers complete composability. Users can define distinct components for data storage, batch generation, pre-storage and post-sampling transforms, allowing independent customization of each aspect.

The RB class has a default constructor that creates a buffer with a generic configuration, utilizing a versatile list storage and a sampler that generates batches uniformly. However, users can also specify the storage to be contiguous in either virtual or physical memory. This format provides faster performance and can handle data sizes up to terabytes. These features allow the creation of buffers that can be populated on the fly during training or used with static datasets for offline RL. TorchRL offers a range of downloadable datasets for this purpose (such as D4RL (Fu et al., 2020) or OpenML (Vanschoren et al., 2013)). Alternative samplers are also available, including a sampler without a replacement that ensures all data is presented once before repeating, or a C++ optimized prioritized sampler. RBs can store any Python object in their native form using the default constructor. Nevertheless, we encourage presenting the data in a TensorDict format, which streamlines the workflow (see Figure 1) and benefits from TorchRL's optimized storage, sampling and transform techniques. TorchRL also integrates a remote replay buffer component to gather data from distributed workers. This functionality is detailed in Appendix D. .

**Modules and models.** In RL, the model architectures are distinct not only from those in other machine learning disciplines but also within RL itself. This creates two interconnected issues: Firstly, there is considerable architectural variability to contend with. Secondly, the nature of inputs and

outputs for these networks can fluctuate based on the specific environment and algorithm in use. Thus, both challenges necessitate a flexible and adaptable approach.

TorchRL responds to the first problem by providing RL-dedicated neural network architectures, which are organically integrated into PyTorch as native `nn.Modules`. These are available at varying levels of abstraction, ranging from foundational building blocks like Multilayer Perceptron (MLP), Convolutional Neural Networks (CNN), Long short-term memory (LSTM) modules and Transformer, to comprehensive, high-level structures such as `ActorCriticOperator` or `WorldModelWrapper` but also exploration modules like `NoisyLinear` or Planners like `CEMPlanner`.

To tackle the second challenge, TorchRL uses specialized `TensorDictModule` and `TensorDictSequential` primitives subclassed from the TensorDict library (see Figure 6). `TensorDictModule` wraps PyTorch modules, transforming them into tensordict-compatible objects for effortless integration into the TorchRL framework. Concurrently, `TensorDictSequential` concatenates TensorDictModules, functioning similarly to `nn.Sequential`. Importantly, these TensorDict modules maintain full compatibility with torch 2.0, torch.compile, and functorch. Vectorized maps (vmap) are also well-supported, enabling the execution of multiple value networks in a vectorized manner. All instances of TensorDictModule are functional and stateful, permitting parameters to be optionally inputted without additional transformations. This facilitates the design of meta-RL algorithms within the TorchRL library.

**Objectives and value estimators.** In TorchRL, objective classes are stateful components that track trainable parameters from `torch.nn` models and handle loss computation, a crucial step for model optimization in any machine learning algorithm. These components share a common signature defined by two main methods: a constructor method that accepts the models and all relevant hyperparameters, and a forward computation method that takes a TensorDict of collected data samples as input and returns another TensorDict with one or more loss terms. Loss modules can accommodate any `nn.Module` and process inputs that are not presented as TensorDict as it is shown in Appendix E.

This design choice makes objective classes very versatile components, abstracting the implementation of a vast array of loss functions derived from various families of RL algorithms using a single template. These include on-policy algorithms like Advantage Actor-Critic (A2C) (Mnih et al., 2016) or Proximal Policy Optimization (PPO) (Schulman et al., 2017), off-policy algorithms like Deep Q-Network (DQN) (Mnih et al., 2013), distributional DQN (Bellemare et al., 2017) and subsequent improvements of these (Hessel et al., 2017), Deep Deterministic Policy Gradient (DDPG) (Lillicrap et al., 2015), Twin Delayed Deep Deterministic Policy Gradient (TD3) (Fujimoto et al., 2018a), Soft Actor-Critic (SAC) (Haarnoja et al., 2018) or Randomized Ensembled Double Q-learning (REDQ) (Chen et al., 2021b), offline algorithms like Implicit Q-Learning (Kostrikov et al., 2021), Decision Transformer (Chen et al., 2021a)), and model-based algorithms like Dreamer (Hafner et al., 2019).

Finally, TorchRL also provides a solution for state value estimations, a crucial step in the loss computation of many RL algorithms that can take various forms. As for the modules, the value estimators are presented both in a functional, explicit way as well as an encapsulated TensorDictModule class that facilitates their integration in TorchRL-based scripts. These replaceable objects, named `ValueEstimators`, can be called outside the objective class or dynamically during the objective forward call, providing a versatile solution to accommodate different forms of loss computation.

## 3 ECOSYSTEM

**Documentation, knowledge-base, engineering.** TorchRL has a rich, near-complete documentation: each public class and function must have a proper set of docstrings. At the time of writing, the coverage is above 90%, and tested across Linux, MacOs and Windows platforms. All the optional dependencies (eg, environment backends or loggers) are being tested in dedicated workflows. Compatibility with all versions of Gym is guaranteed starting from v0.13, including the latest transition to Gymnasium. If both libraries are present in the virtual environment, a dedicated set of utilities can be used to control which backend to pick. Several runtime benchmarks have been put in place to ensure that our modules keep a high-efficiency standard. The ecosystem also includes datasets, issues-tracking, model library and code examples. External forums, such as the PyTorch forum, are monitored for issues related to the library.

**Take-up.** TorchRL has seen rapid growth since its initial open-sourcing. At the time of writing, the library has more than 120 collaborators, is close to reaching 1500 stars on GitHub and has been adopted by many teams of researchers and practitioners. In Appendix J , we express our gratitude to the contributors who have helped in the development and enhancement of TorchRL.

**Applications.** TorchRL is a versatile library that prioritizes comprehensive coverage, striving to address a diverse array of decision-making scenarios. As demonstrated previously, its components exhibit remarkable adaptability, allowing it to fulfill its objective. Fine-tuned, simple examples of popular algorithms testify of the library's versatility. Illustrated through finely-tuned, straightforward examples of popular algorithms, the library's versatility becomes evident.

The scope of coverage extends from off-policy model-free RL (eg, DQN (Mnih et al., 2013), Rainbow (Hessel et al., 2017), SAC (Haarnoja et al., 2018), DDPG (Lillicrap et al., 2015), TD3 (Fujimoto et al., 2018b)) to on-policy model-free RL (PPO (Schulman et al., 2017), A2C (Mnih et al., 2016)). Offline RL, which has garnered significant attention, is also exemplified with algorithms such as IQL (Tan, 1993) and CQL (Kumar et al., 2020), Decision Transformer (Chen et al., 2021a) and Online-Decision Transformer (Zheng et al., 2022), which serve as valuable resources to assist users in developing their own solutions. We ensure compatibility with model-based algorithms through a state-of-the-art implementation of Dreamer (Hafner et al., 2019), which contrasts with most other libraries that typically only provide either model-free or model-based algorithms in isolation. The majority of TorchRL's components seamlessly integrate with Multi-Agent Reinforcement Learning (MARL) paradigms, and the library also offers specialized MARL components. The extent of our solution spectrum can be gauged through the rigorous benchmarking of six MARL algorithms in Appendix H. The library's primitives are also fit for training on distributed settings, as our implementation and results of IMPALA Espeholt et al. (2018) show.

Finally, RL from Human Feedback (RLHF) (Nakano et al., 2021) is also a feature of the library. Several modules and helper functions make it easy to interact with third party libraries such as Hugging-Face transformers (Wolf et al., 2020) or datasets (Lhoest et al., 2021) to fine-tune generative models with little effort. These examples showcase the wide applicability and usefulness of TorchRL and its underlying components. TorchRL is currently used by some research groups on hardware.

## 4 RESULTS

In this section, we experimentally showcase some of our library's key features. We focus on demonstrating code reliability, scalability, flexibility in supporting multiple decision-making paradigms, and component efficiency. In the repository, we make available all scripts used in this section.

**Online single-agent RL.** The most common use case covered by decision making libraries in online RL. We conduct experiments using our components to reproduce well-known results for A2C (Mnih et al., 2016), PPO (Schulman et al., 2017), DDPG (Lillicrap et al., 2015), TD3 (Fujimoto et al., 2018b), and SAC (Haarnoja et al., 2018) algorithms. These experiments are reported in Table 1. In each case, we closely follow the original implementations (including network architectures, hyperparameters and number of training steps) and obtain results that match those of their original papers.

Our training times, with implementations designed mainly to reproduce established benchmarks, including the architectures, are similar to those of other libraries when using a modern desktop GPU. For MuJoCo experiments, on-policy algorithms usually complete in about 1 hour, while off-policy algorithms typically range from 5 to 7 hours. In the case of Atari experiments, the training time is approximately 8 hours. Note that TorchRL offers additional options to accelerate training times, as can be seen for example in Table 5, which we did not use to maintain fidelity with the original works.

**Distributed RL.** TorchRL's distributed components enable the replication of scalable methods. As an illustration, we provide results for the IMPALA (Espeholt et al., 2018) algorithm, which utilizes distributed data collection and a centralized learner. Our implementation adheres to the original paper's specifications, and we have validated it across several Atari environments, as shown also in Table 1. More details as well as training plots are provided in subsection G.1.

**Offline RL.** TorchRL also includes support for offline methods. In Table 2, results are available for implicit Q-Learning (IQL) (Kostrikov et al., 2021), the Decision Transformer (DT) (Chen et al., 2021a)

Table 1: Experimental training of multiple on-policy and off-policy algorithms. We run each training 5 times with different seeds and report the mean final reward and std.

| | HalfCheetah[†] | Hopper[†] | Walker2D[†] | Ant[†] | Pong[‡] | Freeway[‡] | Boxing[‡] | Breakout[‡] |
|---|---|---|---|---|---|---|---|---|
| A2C | 836±964 | 493±192 | 381±109 | 54±20 | 20.57±0.65 | 30.58±1.40 | 88.76±6.45 | 375.65±47.34 |
| PPO [*] | 2770±821 | 1703±742 | 3336±928 | 2469±606 | 20.52±0.58 | 33.31±0.95 | 98.31±0.93 | 335.71±46.72 |
| DDPG | 10433±357 | 914±144 | 1683±761 | 1169±658 | - | - | - | - |
| TD3 | 10285±837 | 2809±618 | 4409±256 | 5373±165 | - | - | - | - |
| SAC | 11077±323 | 2963±644 | 4561±92 | 3467±654 | - | - | - | - |
| IMPALA | - | - | - | - | 20.54±0.18 | 0.0±0.0 | 99.19±0.54 | 525.57±105.47 |

[*] Our implementations compute the Generalized Advantage Estimator (GAE) at every epoch.
[†] MuJoCo environments v3, trained for 1M steps.
[‡] Atari 2600 environment, trained for 40M frames (10M steps) for A2C and PPO and 200M for IMPALA.

Table 2: Experimental training of offline algorithms. We run each training 5 times with different seeds and report the mean final reward and std.

| | HalfCheetah[†] | Hopper[†] | | | HalfCheetah[†] | Hopper[†] | | | HalfCheetah[†] | Hopper[†] |
|---|---|---|---|---|---|---|---|---|---|---|
| DT | 4916±30 | 1048±226 | | oDT | 4968±58 | 2830±96 | | IQL | 4864±147 | 1418±131 |

[†] D4RL offline RL (medium replay) datasets, trained for 50000 gradient updates.

and the Online Decision Transformer (oDT) (Zheng et al., 2022) on 2 MuJoCo environments, with all three implemented following their respective original papers. Plots are available in subsection G.2.

**Multi-agent.** To bootstrap the adoption of TorchRL for multi-agent use cases, we provide implementations for several state-of-the-art MARL algorithms and benchmark them on three multi-robot coordination scenarios in the VMAS (Bettini et al., 2022) simulator. The results, reported in Figure 2, show the correctness of TorchRL's implementations, matching the results from (Bettini et al., 2022). In the repository, in addition to making available the scripts used in this evaluation, we provide examples and tutorials to illustrate how TorchRL can be seamlessly used in MARL contexts. Further details on MARL experiments and comparisons with other libraries are available in Appendix H.

**Vectorized data collection.** The philosophy of TorchRL and its core component, TensorDict, enables seamless compatibility with vectorized (batched) simulators (such as IsaacGym (Makoviychuk et al., 2021), Brax (Freeman et al., 2021), and VMAS (Bettini et al., 2022)). These vectorized simulators run parallel environment instances in a batch, leveraging the SIMD paradigm of GPUs to greatly speed up execution. While this paradigm is becoming popular thanks to advances in hardware, not all RL libraries are ready to leverage its benefits. For instance, in Table 3 we report a comparison between TorchRL and RLlib (Liang et al., 2018) showing the collected frames per second as a function of the number of vectorized environments for the "simple spread" task in the VMAS simulator. The comparison shows that TorchRL is able to leverage vectorization to greatly increase collected frames. It serves to illustrate the usage of hardware accelerators in simulators, such as IsaacGym, which is also covered by the library's wrappers. The code used to generate the results is publicly available in the benchmark section of our repository.

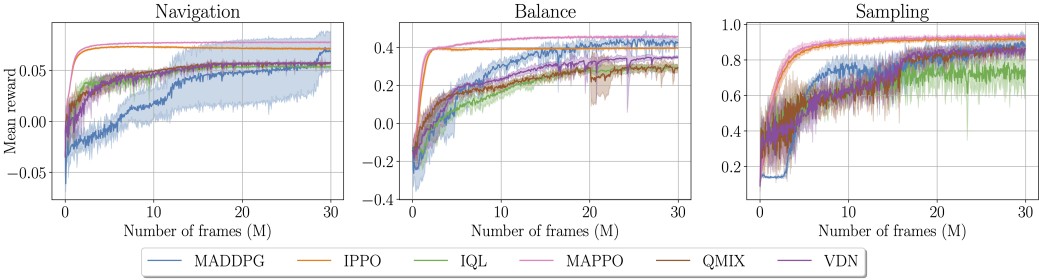

Figure 2: MARL algorithm evaluation in three VMAS multi-robot control tasks. We report the mean and standard deviation reward over 6 random seeds. Each run consists of 500 iterations of 60, 000 steps each, with an episode length of 100 steps.

Table 3: Frames per second as a function of the number of vectorized environments. This evaluation shows that TorchRL is able to leverage vectorized simulators to increase collected frames. The experiments were run on 100 steps of the "simple spread" MPE (Lowe et al., 2017) scenario in the VMAS (Bettini et al., 2022) vectorized simulator.

| # of vectorized envs | 1 | 3334 | 6667 | 10000 | 13333 | 16667 | 20000 | 23333 | 26666 | 30000 |
|---|---|---|---|---|---|---|---|---|---|---|
| TorchRL fps | 127 | 263848 | 398304 | 529872 | 444088 | 522537 | 538930 | 487261 | 384469 | 422003 |
| RLlib fps | 93 | 1885 | 1663 | 1675 | 1580 | 1523 | 1581 | 1447 | 1373 | 1336 |

**Computational efficiency.** To check that our solution does not come at the cost of reduced throughput, we test two selected functionalities (advantage computation and data collection) against other popular libraries. These experiments were run on an AWS cluster with a single node (96 CPU cores and 1 A100 GPUs per run). These results are reported in Table 4 and Table 5. Overall, these results show that our solutions have a comparable if not better throughput than others.

Table 4: Efficiency computing the Generalised Advantage Estimation (GAE) (Schulman et al., 2015). The data consisted of a batch of 1000 trajectories of 1000 time steps, with a "done" frequency of 0.001, randomly spread across the data.
† Using these implementations requires transforming tensors to Numpy arrays and then transforming the result back to tensors. Numpy and similar backends require moving data from and to GPU which can substantially impact performance. Those operations are unaccounted for in this table but can double the GAE runtime. On the contrary, TorchRL works on PyTorch tensors directly. * A proper usage of Ray's GAE implementation would have needed a split of the adjacent trajectories, which we did not do to focus on the implementation efficiency.

| Library | Speed (ms) | Standalone | Meta-RL | Adjacent | Backend | Device |
|---|---|---|---|---|---|---|
| Tianshou† | 5.15 | - | - | + | namba | CPU |
| SB3† | 14.1 | - | - | + | numpy | CPU |
| RLLib (Ray)* | 9.38 | - | - | - | scipy | CPU |
| CleanRL | 1.43 | - | - | + | Jax | GPU |
| TorchRL-compiled | 2.67 | + | + | + | PyTorch | GPU |
| TorchRL-vec | **1.33** | + | + | + | PyTorch | GPU |

Table 5: Data collection speed with common gym environments across common RL libraries. In each experiment, 32 workers were used. The scripts are available on TorchRL's benchmark folder.

| Library | Breakout-v5 | HalfCheetah-v4 | Pendulum-v1 |
|---|---|---|---|
| Tianshou | 1212 | 4719 | 5823 |
| RLLib | 97 | 1868 | 1845 |
| Gymnasium | 9289 | 24388 | 25910 |
| TorchRL (parallel env) | 9092 | 21742 | 23363 |
| TorchRL (sync collector) | 10394 | 23527 | 24530 |
| TorchRL (async collector) | **19401** | **32894** | **31584** |

## 5 CONCLUSION

TorchRL is a modular decision-making library with composable, single-responsibility building pieces that rely on TensorDict for an efficient class-to-class communication protocol. TorchRL's design principles are tailored to the dynamic field of decision-making, enabling the implementation of a wide range of algorithms. We believe TorchRL can be of interest to researchers and developers alike. Its key advantage is that researchers can create or extend existing classes on their own and seamlessly integrate them with existing scripts without the need to modify the internal code of the library, allowing for quick prototyping while maintaining a stable and reliable code base. We believe this feature will also be suited for developers looking to apply an AI solution to a specific problem. We demonstrate that TorchRL benefits from a wide range of functionalities, is reliable and exhibits computational efficiency. The library is fully functional, and tested across various scenarios and applications.

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

APPENDIX

## A    OPENSOURCE REPOSITORIES

TorchRL and TensorDict libraries opensource code can be found at the following locations:

- TorchRL: `https://github.com/pytorch/rl`
- TensorDict: `https://github.com/pytorch/tensordict`

## B    TENSORDICT FUNCTIONALITY

TensorDict's aim is to abstract away functional details of PyTorch workflows, enabling scripts to be more easily reused across different tasks, architectures, or implementation details. While primarily a tool for convenience, TensorDict can also provide notable computational benefits for specific operations.

### B.1    TENSORDICTBASE AND SUBCLASSES

The parent class of any TensorDict class is `TensorDictBase`. This abstract class lists the common operations of all its subclasses and also implements a few of them. Indeed, some generic methods are implemented once for all subclasses, such as `tensordict.apply`, as the output will always be a `TensorDict` instance. Others, such as `__getitem__` have a behavior that is intrinsically linked to their specifics.

The following subclasses are available:

- `TensorDict`: this is the most common class and usually the only one users will need to interact with.
- `LazyStackedTensorDict`: This class is the result of calling `torch.stack` on a list of `TensorDictBase` subclass instances. The resulting object stores each instance independently and stacks the items only when queried through `__getitem__`, `__setitem__`, get, `set` or similar. Heterogeneous TensorDict instances can be stacked together: in that case, `get` may fail to stack the tensors (if one is missing from one tensordict instance or if the shapes do not match). However, this class can still be used to carry data from object to object or worker to worker, even if the data is heterogeneous. The original tensordicts can be recovered simply via indexing of the lazy stack: `torch.stack(list_of_tds, 0)[0]` will return the first element of `list_of_tds`.
- `PersitentTensorDict`: implements a TensorDict class with persistent storage. At the time of writing, only the H5 backend is implemented. This interface is currently being used to integrate massive datasets within TorchRL for offline RL and imitation learning.
- `_CustomOpTensorDict`: This abstract class is the parent of other classes that implement lazy operations. This is used to temporarily interact with a TensorDict on which a zero-copy reshape operation is to be executed. Any in-place operations on these instances will affect the parent tensordict as well. For instance, `tensordict.permute(0, 1).set("a", tensor)` will set a new key in the parent `tensordict` as well.

**Performance**    Through a convenient and intuitive API, tensordict offers some out-of-the-box optimizations that would otherwise be cumbersome to achieve and, we believe, can have a positive impact beyond RL usage. For instance, TensorDict makes it easy to store large datasets in contiguous physical memory through a PyTorch interface with NumPy's memory-mapped arrays named `MemmapTensor`, available in the tensordict library. Unlike regular PyTorch datasets, TensorDict offers the possibility to index multiple items at a time and make them immediately available in memory in a contiguous manner: by reducing the I/O overhead associated with reading single files (which we preprocess and store in a memory-mapped array on disk), one can read, cast to GPU and preprocess multiple files all-at-once. Not only can we amortize the time of reading independent files, but we can also leverage this to apply transforms on batches of data instead of one item at a time. These batched transforms on device are much faster to execute than their multiprocessed

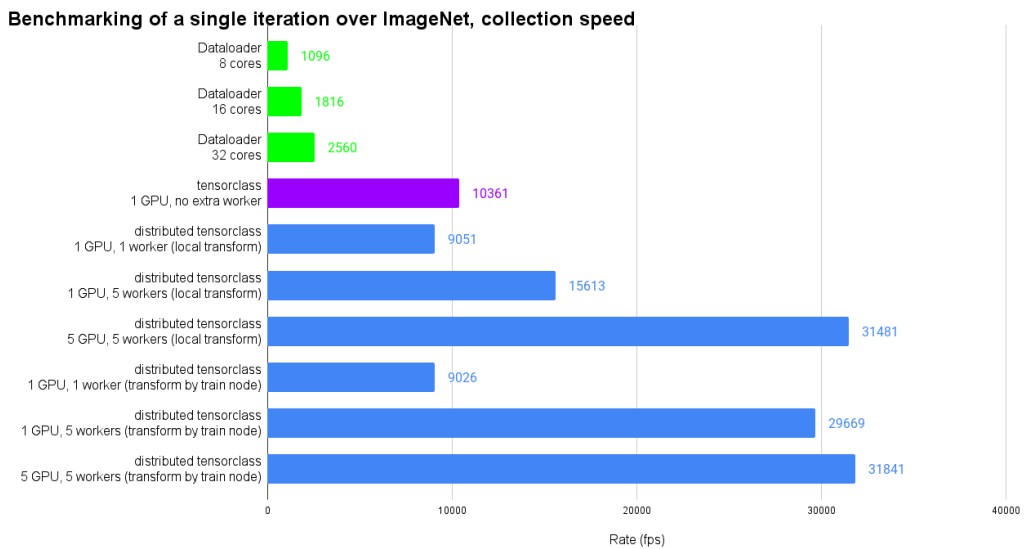

Figure 3: Dataloading speed with TensorDict.

```python
@tensorclass
class MyClass:
    image: torch.Tensor
    label: torch.Tensor

data = MyClass(
    image=torch.zeros(100, 32, 32, 3, dtype=torch.uint8),
    label=torch.randint(1000, (100,)),
    batch_size=[100]
)

# a MyData instance with floating point content
data_float = data.apply(lambda x: x.float())
# a MyData instance on cuda
data_cuda = data.to("cuda")
# the first 10 elements of the original MyData instance
data_idx = data[:10]
```

Figure 4: A @tensorclass example.

counterparts. Figure 3 shows the collection speed over ImageNet with batches of 128 images and per-image random transformations including cropping and flipping. The workflow used to achieve this speedup is available in the repository documentation.

**tensorclass** TensorDict also offers a @tensorclass decorator aimed at working as the @dataclass decorator: it allows for the creation of specialized dataclasses with all the features of TensorDict: shape vs key (attribute) dimension, device, tensor operations and more. @tensorclass instances natively support any TensorDict operation. Figure 4 shows a few examples of @tensorclass usage.

## B.2 TENSORDICT.NN: MODULES AND FUNCTIONAL API

The second pane of the tensordict library is its module interface, on which TorchRL heavily relies. tensordict.nn aims to address two core limitations of torch.nn. First, torch.nn offers a Module class which encourages users to create new components through inheritance. However, in many cases, this level of control in module construction is unnecessary and can even hinder

```python
class MLP(nn.Module):
    def __init__(self):
        super().__init__()
        self.layer1 = nn.Linear(3, 4)
        self.layer2 = nn.Linear(4, 2)
        self.activation = torch.relu

    def forward(self, x):
        y = self.activation(self.linear1(x))
        return self.linear2(y)
```

Figure 5: Programmable module design with TensorDictModule

```python
from tensordict.nn import TensorDictSequential as Seq, TensorDictModule
                                  as Mod
module = Seq(
   Mod(nn.Linear(3, 4), in_keys=['input'], out_keys=['hidden0']),
   Mod(torch.relu, in_keys=['hidden0'], out_keys=['hidden1']),
   Mod(nn.Linear(4, 2), in_keys=['hidden1'], out_keys=['output']),
)
```

Figure 6: Programmable module design with TensorDictModule

modularity. For example, consider the coding of a simple multi-layered perceptron, as depicted in figure 5, which is commonly encountered.

Such a definition of a dedicated class lacks flexibility. One solution commonly adopted to control the network hyperparameters (number of layers, the activation function, or layer width) is to pass them as inputs to the constructor and group the layers together within a `nn.Sequential` instance. This solution works fine as long as the sequence is a simple chain of operations on a single tensor. As soon as some operations need to be skipped, or when modules output multiple tensors, `nn.Sequential` is not suited anymore and the operations must be explicitly written within the `forward` method, which sets the computational graph once and for all. The `TensorDictModule` class overcomes this limitation and provides a fully programmable module design API, as shown in figure 6.

Formulating a module in this way brings all the flexibility we need to address the shortcomings exposed above: it is easy to programmatically design a new model architecture at runtime for any kind of model. Selecting subgraphs is easy and can be achieved via regular indexing of the module or more advanced sub-graph selection. In the following example, a call to `select_subsequence` will output a new `TensorDictSequential` where the modules are restricted only to those whose operations depend on the `"hidden1"` key:

```python
sub_module = module.select_subsequence(in_keys=["hidden1"])
```

To further back our claim that such model construction is beneficial both in terms of clarity and modularity, let us take the example of writing a residual connection with tensordict.nn:

```python
block = Seq(
  Mod(nn.Linear(128, 128), in_keys=['x'], out_keys=['intermediate']),
  Mod(torch.relu, in_keys=['intermediate'], out_keys=['intermediate']),
  Mod(nn.Linear(128, 128), in_keys=['intermediate'], out_keys=['
                                  intermediate']),
)
residual = Seq(
  block,
  Mod(lambda x, z: x+z, in_keys=['x', 'intermediate'], out_keys=['x'])
)
```

For clarity, we have separated the backbone and the residual connection, but these could be grouped under the same `TensorDictSequential` instance. Nevertheless, `select_subsequence` will go through these various levels of nesting if ever called.

**Functional usage**  Functional usage of these modules is a cornerstone in TorchRL as it underpins efficient calls to the same module with different sets of parameters for off-policy algorithms where multiple critics are executed simultaneously or in situations where target parameters are needed. `tensordict.nn` provides a `make_functional` function that transforms any `torch.nn` module in a functional module that accepts an optional `params` (keyword-)argument. This function will return the parameters and buffers in a `TensorDict` instance whose nested structure mimics the one of the originating module (unlike `Module.state_dict()` which returns them in a flat structure).

Using these functionalized modules is straightforward. For instance, we can zero all the parameters of a module and call it on some data:

```
params = make_functional(module)
params_zero = params.clone().zero_()
module(tensordict, params=params_zero)
```

This allows us to group the calls to multiple critics in a vectorized fashion. In the following example, we stack the parameters of two critic networks and call the module with this stack using `torch.vmap`, the PyTorch vectorized-map function:

```
make_functional(critic)
vmap_critic = vmap(critic, (None, 0))
data = vmap_critic(
  data,
  torch.stack([params_critic0, params_critic1], 0))
```

We refer to the `torch.vmap` documentation for further information on this feature. Similarly, calling the same module with trainable or target parameters can be done via a functional call to it.

**Performance**  The read and write operations executed during the unfolding of a `TensorDictSequential` can potentially impact the execution speed of the underlying model. To address this concern, we offer a dedicated `symbolic_trace` function that simplifies the operation graph to its essential elements. This function generates a module that is fully compatible with `torch.compile`, resulting in faster execution. By combining these solutions, we can achieve module performance that is on par, if not better, than their regular `nn.Module` counterparts.

**Using TensorDict modules in tensordict-free code bases**  Because the library's goal is to be as minimally restrictive as possible, we also provide the possibility of executing TensorDict-based modules without explicitly requiring any interaction with TensorDict: the `tensordict.nn.dispatch` decorator allows to interact with any module from the TensorDict ecosystem with pure tensors:

```
module = TensorDictModule(
  lambda x: x+1, x-2,
  in_keys=["x"], out_keys=["y", "z"])
y, z = module(x = torch.randn(3))
```

`TensorDictModule` and associated classes also provide a `select_out_keys` method that allows to hide some specific keys of a graph to minimize the output. When used in conjunction with `@dispatch`, this enables a much clearer usage when multiple intermediate keys are present.

## B.3 TENSORDICT VS OTHER TENSOR CARRIERS

TensorDict is one of the many data carriers in RL and other machine learning fields. To avoid overfitting to a single comparison, we provide a generic comparison of TensorDict in the RL ecosystem followed by a comparison of TensorDict with PyTrees.

### B.3.1 TENSORDICT VS OTHER RL DATA CARRIERS

Examples of existing data carriers in RL are Tianshou's Batch, RLLib's TensorDict, or Salina's Workspace. The goal of each of these classes is somewhat similar, i.e., abstracting away the precise content of a module output to focus on the orchestration of the various components of the library.

First and foremost, TensorDict isn't focused on RL specifically, while the others are. We did not restrict its content to be some derivation of a (`observation`, `next_observation`, `reward`, `done`, `action`) tuple. It can store any numerical data, e.g., structures without reward for imitation learning, or with deeply nested data structures for Monte Carlo Tree Search algorithms. This allows TorchRL to cover a much broader range of algorithms like Offline RL, imitation learning, optimal control and others, where other solutions may struggle given their content limitations.

Second, TensorDict behaves like a tensor: it supports shape manipulation, casting, copying, etc. All these operations are implemented in such a way that in many places, tensors and TensorDicts are interchangeable. This allows us to build policies that return TensorDicts and not tensors for compound actors for instance. It also allows for a certain implicit recursivity in TensorDict functionalities: for many operations, applying a transformation to all leaves in the TensorDict tree just boils down to calling that operation over all fields of the TensorDict, recursively.

### B.3.2 TENSORDICT VS PYTREE

TensoDict can also be compared to PyTree, a concept shared by PyTorch and Jax that allows to cast operations to any level of deeply nested data structures. Yet, TensorDict isn't based on PyTree. There are several things to take into consideration to understand why:

- Using PyTree implies decomposing the TensorDict, applying the function to its tensors (leaves), and recomposing it. This introduces some overhead, as metadata need to be saved, and the class reinstantiated which does not come for free. 'TensorDict.apply' can reduce that overhead by knowing in advance what to do with the op that is applied to it.

- TensorDict operations are implemented in a way that makes the results more directly useful to the user. Splitting (or unbinding) operations are a good example of this: using a regular PyTree call over a nested dictionary would result in nested dictionaries of tuples of tensors. Using 'TensorDict.split' will result in tuples of TensorDicts, which matches the 'torch.split' signature. This makes this operation more "intuitive" than the PyTree version and is more directly useful to the user. Of course, some PyTree implementations offers tools to invert the structure order. Nevertheless, we believe having these features built-in to be useful.

- TensorDicts carry important metadata, like "shape" (i.e., batch-size) and "device", whereas generic structures that are compatible with PyTree (such as dict and lists) do not carry this information. In a sense, it is less generic than a dictionary, but for its own good. Metadata allow us to leverage the concept of batch-size for instance, and separate it from the "feature-size". Among other advantages, this greatly facilitates vmap-ing over a TensorDict, as we know what shape can be vmaped over in advance. For example, TensorDict used in TorchRL have by convention the "time" dimension positioned last. If we need to vmap over it, we can call `vamp(op, (-1,))(data)`. This `-1` is generic (i.e., reusable across use cases, independently of the number of dimensions), and does not depend on the batch size. This is anecdotal example would be harder to code using PyTrees.

- We think that using PyTree requires more knowledge and practice than TensorDict. Sugar code like `data.cuda()` would be harder to code with PyTree. Retrieving the resulting metadata (through `data.device`) wouldn't be possible either, since PyTree does not carry metadata unless a special class is written.

- TensorDicts come with some handy features that would not be easy to come up with using PyTrees: they can be locked to avoid unwanted changes and they can be used as context managers for some operations (like functional calls over modules, or temporarily unlocking a tensordict using `with data.unlock_():...`).

- Finally, note that TensorDict is registered within PyTree in PyTorch, such that anything PyTree can do, TensorDict can do too.

### B.4    TENSORDICT OVERHEAD MEASUREMENT

We measured TensorDict overhead for the lastest release at the time of writing the manuscript (v0.2.1) against other existing solutions.

#### B.4.1    TENSORDICT VS PYTREE

TensorDict has some similarities with the PyTree data-structure presented in Jax Bradbury et al. (2018) or in PyTorch Paszke et al. (2019), as it can dispatch operations over a set of leaves. To measure the worst case overhead introduced by TensorDict (which handles meta-data such as batch-size and carrier device when executing these operations), we executed the same operations using PyTree and TensorDict over a highly nested (100 levels) dictionary, where each leaf was a scalar tensor. We consider this level of nesting to be above most of the usual use cases of TensorDict (in a typical RL setting, only a couple of levels are needed for data representation and a tenth or so for parameter representation with complex models). We tested the time it takes to increase the value of each leaf by 1.0 or to transfer all tensors from CPU to a GPU device. Additionally, we also assessed the time required to split, chunk, and unbind a TensorDict along its batch-size compared to the equivalent PyTree operation. We ran these operations 1000 times over 16 different runs and report the 95% confidence interval of the expected value.

**Results** The value increment took 0.916 ms (0.91531-0.91628) with PyTree, compared with 1.0555 ms (1.05495 - 1.05608) for TensorDict, which constitutes a 115% runtime increment. Casting to CUDA had a similar impact, with PyTree running at 1.181 ms (1.16242 - 1.19960) and TensorDict at 1.302 ms (1.30065 - 1.30242), a 10% runtime increment. For batch-size operations, the overhead was also worse for TensorDict: split took 1.015 ms (1.01436 - 1.01530) against 4.254 ms (4.23771 - 4.26700), chunk took 0.801 ms (0.80027 - 0.80084) vs 2.155 ms (2.14920 - 2.16060) and unbind 1.418 ms (1.417299 - 1.41778) vs 3.836 ms (3.834133 - 3.83615) for PyTree and TensorDict in each case. Recall that those operations with PyTree returned nested dictionaries of tuples rather than tuples of dictionaries and that TensorDict had to re-compute the batch-size of each element of the batch: these differences explain the overhead observed. Nevertheless, we will aim at addressing these issues in the next TensorDict release as there is significant room for improvement. Our plan is detailed on the discussion page of the repository.

#### B.4.2    TENSORDICT FUNCTIONAL CALLS VS FUNCTORCH

TensorDict offers a simple API to execute functional calls over a specific PyTorch module. Unlike `torch.func.functinoal_call`, TensorDict allows for functional calls over any method of the module, and its nested representation makes it possible to execute functional calls over sub-modules without effort. For the sake of completeness, we also compared how much a functional call over a narrow (16 cells instead of 2048) but deep `torch.nn.Transformer` would be impacted by a switch to TensorDict. The runtime for functorch was of 8.423 ms (8.38470 - 8.46124) against 10.26071 ms (10.25635 - 10.26500) for TensorDict. This reduced speed should be put in perspective with the depth of the model and reduced runtime for the algebraic operations. Note that TensorDict functional calls are currently in their first version and will be refactored soon to make them easier to use and more broadly applicable (eg, in distributed and graph settings). We have already developed a prototype of this feature which will be included in the next release. After this refactoring, early experiments have already confirmed that the runtime will be at par with existing functional calls in functorch.

## C    ENVIRONMENT API

To bind input/output data to a specific domain, TorchRL uses `TensorSpec`, a custom version of the feature spaces found in other environment libraries. This class is specially tailored for PyTorch: its instances can be moved from device to device, reshaped, or stacked at will, as the tensors would be. TensorDict is naturally blended within `TensorSpec` through a `CompositeSpec` primitive which is a metadata version of TensorDict. The following example shows the equivalence between these two classes:

```
>>> from torchrl.data import UnboundedContinuousTensorSpec, \ ...
```

```
    CompositeSpec
>>> spec = CompositeSpec(
...     obs=UnboundedContinuousTensorSpec(shape=(3, 4)),
...     shape=(3,),
...     device='cpu'
)
>>> print(spec.rand())
TensorDict(
    fields={
        obs: Tensor(shape=torch.Size([3, 4]), device=cpu, dtype=torch.
                                        float32, is_shared=False)},
    batch_size=torch.Size([3]),
    device=cpu,
    is_shared=False)
```

`TensorSpec` comes with multiple dedicated methods to run common checks and operations, such as `spec.is_in` which checks if a tensor belongs to the spec's domain, `spec.project` which maps a tensor onto its L1 closest data point within a spec's domain or `spec.encode` which creates a tensor from a NumPy array. Other dedicated operations such as conversions from categorical encoding to one-hot are also supported.

Specs also support indexing and stacking, and as for TensorDict, heterogeneous `CompositeSpec` instances can lazily be stacked together for better integration in multitask or multiagent frameworks.

Using these specs greatly empowers the library and is a key element of many of the performance optimizations in TorchRL. For instance, using the TensorSpecs allows us to preallocate buffers in shared memory or on GPU to maximize the throughput during parallel data collection, without executing the environment operations a single time. For real-world deployment of TorchRL's solutions, these tools are essential as they allow us to run algorithms on fake data defined by the environment specs, checking that there is no device or shape mismatch between model and environment, without requiring a single data collection on hardware.

**Designing environments in TorchRL**   To encode a new environment, developers only need to implement the internal `_reset()` and `_step()` methods and provide the input and output domains through the specs for grounding the data. TorchRL provides a set of safety checks grouped under the `torchrl.envs.utils.check_env_specs` which should be run before executing an environment for the first time. This optional validation step offloads many checks that would otherwise be executed at runtime to a single initial function call, thereby reducing the footprint of these checks.

## D  REPLAY BUFFERS

TorchRL's replay buffers are fully composable: although they come with "batteries included", requiring a minimal effort to be built, they also support many customizations such as storage type, sampling strategy or data transforms. We provide dedicated tutorials and documentation on their usage in the library. The main features can be listed as follows:

• Various **Storage** types are proposed. We provide interfaces with regular lists, or contiguous physical or virtual memory storages through MemmapTensor and torch. Tensor classes respectively.

• At the time of writing, the **Samplers** include a generic circular sampler, a sampler without repetition, and an efficient, C++ based prioritized sampler.

• One can also pass **Transforms** which are notably identical to those used in conjunction with TorchRL's environments. This makes it easy to recycle a data transform pipeline used during collection to one used offline to train from a static dataset made of untransformed data. It also allows us to store data more efficiently: as an example, consider the cost of saving images in uint8 format against saving transformed images in floating point numbers. Using the former reduces the memory footprint of the buffer, allowing it to store more data and access it faster. Because the data is stored contiguously and the transforms are applied on-device, the reduction in memory footprint comes at a little extra computational cost. One could also consider the usage of transforms

```python
from torchrl.objectives import DQNLoss
from torchrl.data import OneHotDiscreteTensorSpec
from torch import nn
import torch

n_obs = 3
n_action = 4

action_spec = OneHotDiscreteTensorSpec(n_action)
value_network = nn.Linear(n_obs, n_action) # a simple value model
dqn_loss = DQNLoss(value_network, action_space=action_spec)

# define data
observation = torch.randn(n_obs)
next_observation = torch.randn(n_obs)
action = action_spec.rand()
next_reward = torch.randn(1)
next_done = torch.zeros(1, dtype=torch.bool)
# get loss value
loss_val = dqn_loss(
  observation=observation,
  next_observation=next_observation,
  next_reward=next_reward,
  next_done=next_done,
  action=action)
```

Figure 7: TensorDict-free DQN Loss usage.

when working with Decision Transformers (Chen et al., 2021a), which are typically trained with a different lookback window on the collected data than the one used during inference. Duplicating the same transform in the environment and in the buffer with a different set of parameters facilitates the implementation of these techniques.

Importantly, this modularity of the buffer class also makes it easy to design new buffer instances, which we expect to have a positive impact on researchers. For example, a team of researchers developing a novel and more efficient sampling strategy can easily build new replay buffer instances using the proposed parent parent class while focusing on the improvement of a single of its pieces.

**Distributed replay buffers**: our buffer class supports Remote Procedural Control (RPC) calls through `torch.distributed.rpc`. In simple terms, this means that these buffers can be extended or accessed by distant nodes through a remote call to `buffer.extend` or `buffer.sample`. With the utilization of efficient shared memory-mapped storage when possible and multithreaded requests, the distributed replay buffer significantly enhances the transfer speed between workers, increasing it by a factor of three to ten when compared to naive implementations.

# E USING LOSS MODULES WITHOUT TENSORDICT

In an effort to free the library from its bounds to specific design choices, we make it possible to create certain loss modules without any interaction with TensorDict or related classes. For example, figure 7 shows how a DQN loss function can be designed without recurring to a `TensorDictModule` instance.

Although the number of loss modules with this functionality is currently limited, we are actively implementing that feature for a larger set of objectives.

## F TRAINER

The modularity of TorchRL's components allows developers to write training scripts with explicit loops for data collection and neural network optimization in the traditional structure adopted by most ML libraries. While this enables practitioners to keep tight control over the training process, it might hinder TorchRL's adoption by first-time users looking for one-fits-all solutions. For this reason, we also provide a `Trainer` class that abstract this further complexity. By exposing a simple `train()` method, trainers take care of running data collection and optimization while providing several hooks (i.e., callbacks) at different stages of the process. These hooks allow users to customize various aspects of data processing, logging, and other training operations. Trainers also encourage reproducibility by providing a checkpointing interface that allows to abruptly interrupt and restart training at any given time while saving models and RB of any given size.

The trainer executes a nested loop, consisting of an outer loop responsible for data collection and an inner loop that utilizes this data or retrieves data from the replay buffer to train the model. The Trainer class does not constitute a fundamental component; rather, it was developed as a simplified entry point to novice practitioners in the field.

In Figure 8 we provide two code listing examples to train a DDPG agent. The left approach utilizes the high-level Trainer class. On the other hand, the right listing explicitly defines the training loop, giving more control to the user over every step of the process.

## G SINGLE-AGENT REINFORCEMENT LEARNING EXPERIMENTS

To test the correctness and effectiveness of TorchRL framework components, we provide documented validations of 3 online algorithms (PPO, A2C, IMPALA) and 3 offline algorithms (DDPG, TD3 and SAC) for 4 MuJoCo environments (HalfCheetah-v3, Hopper-v3, Walker2d-v3 and Ant-v3) and 4 Atari environments (Pong, Freeway, Boxing and Breakout). Training plots are shown in Figure 9, Figure 11 and Figure 12.

### G.1 ONLINE REINFORCEMENT LEARNING EXPERIMENTS

#### G.1.1 IMPLEMENTATION DETAILS

For A2C (Mnih et al., 2016) and PPO (Schulman et al., 2017), we reproduce the results from the original work on MuJoCo and Atari environments, using the same hyperparameters and network architectures. We also reproduce Atari results for the IMPALA (Espeholt et al., 2018) distributed method. Ensuring a fair comparison among the off-policy algorithms DDPG (Lillicrap et al., 2015), TD3(Fujimoto et al., 2018b), and SAC(Haarnoja et al., 2018), we uniformly apply the same architecture, optimizer, learning rate, soft update parameter, and batch size as used in the official TD3 implementation. Exploration-specific parameters for DDPG and TD3 are taken directly from the according paper. To initiate the process, each algorithm begins with 10,000 steps of random actions, designed to adequately populate the replay buffer.

#### G.1.2 HYPERPARAMETERS

Tables 6, 7 and 8 display all hyperparameters values and network architecture details required to reproduce our online RL results.

### G.2 OFFLINE REINFORCEMENT LEARNING EXPERIMENTS

#### G.2.1 IMPLEMENTATION DETAILS

To replicate the pre-training results of both the Decision Transformer Chen et al. (2021a) and the Online Decision Transformer Zheng et al. (2022), we utilize the base GPT-2 transformer from Hugging Face Wolf et al. (2020), as presented in the official implementations. All parameters concerning architecture and training procedures are adopted directly from the specifications provided in the publications. Correspondingly, for IQL Kostrikov et al. (2021), we have selected the architecture and parameters mentioned in the paper to enable performance reproduction.

```python
# Environment
env = GymEnv('Pendulum-v1')
# Model: Actor and value
mlp_actor = MLP(
    num_cells=64,
    depth=3,
    in_features=3,
    out_features=1
)
actor = TensorDictModule(
    mlp_actor,
    in_keys=['observation'],
    out_keys=['action']
)
mlp_value = MLP(
    num_cells=64, depth=2,
    in_features=4,
    out_features=1
)
critic = TensorDictSequential(
    actor,
    TensorDictModule(
        mlp_actor,
        in_keys = [
            'observation',
            'action',
        ],
        out_keys =
        ['state_action_value']
    )
)
# Data Collector
collector = SyncDataCollector(
    env,
    AdditiveGaussianWrapper(
        actor
    ),
    frames_per_batch=1000,
    total_frames=1000000,
)
# Replay Buffer
buffer = TensorDictReplayBuffer(
    storage=LazyTensorStorage(
        max_size=100000,
    ),
)
# Loss Module
loss_fn = DDPGLoss(
    actor, crititc,
)
optim=torch.optim.Adam(
    loss_fn.parameters(),
    lr=2e-4,
)
# Trainer
Trainer(
    collector=collector
    total_frames=1000000,
    frame_skip=1,
    optim_steps_per_batch=1,
    loss_module=loss_fn,
    optimizer=optim,
)
trainer.train()
```

```python
 # Environment
env = GymEnv('Pendulum-v1')
# Model: Actor and value
mlp_actor = MLP(
    num_cells=64,  depth=3,
    in_features=3,
    out_features=1
)
actor = TensorDictModule(
    mlp_actor,
    in_keys=['observation'],
    out_keys=['action']
)
mlp_value = MLP(
    num_cells=64,
    depth=2,
    in_features=4,
    out_features=1
)
critic = TensorDictSequential(
    actor,
    TensorDictModule(
        mlp_actor,
        in_keys = [
            'observation',
            'action',
        ],
        out_keys =
        ['state_action_value']
    )
)
# Data Collector
collector = SyncDataCollector(
    env,
    AdditiveGaussianWrapper(
        actor
    ),
    frames_per_batch=1000,
    total_frames=1000000,
)
# Replay Buffer
buffer = TensorDictReplayBuffer(
    storage=LazyTensorStorage(
        max_size=100000,
    ),
)
# Loss Module
loss_fn = DDPGLoss(
    actor, crititc,
)
optim=torch.optim.Adam(
    loss_fn.parameters(),
    lr=2e-4,
)
# Training loop
for data in collector:
    buffer.extend(data)
    sample = buffer.sample(50)
    loss = loss_fn(sample)
    loss = loss['loss_actor'] + \
           loss['loss_value']
    loss.backward()
    optim.step()
    optim.zero_grad()
```

Figure 8: Trainer class example usage (left). Fully defined training loop (right).

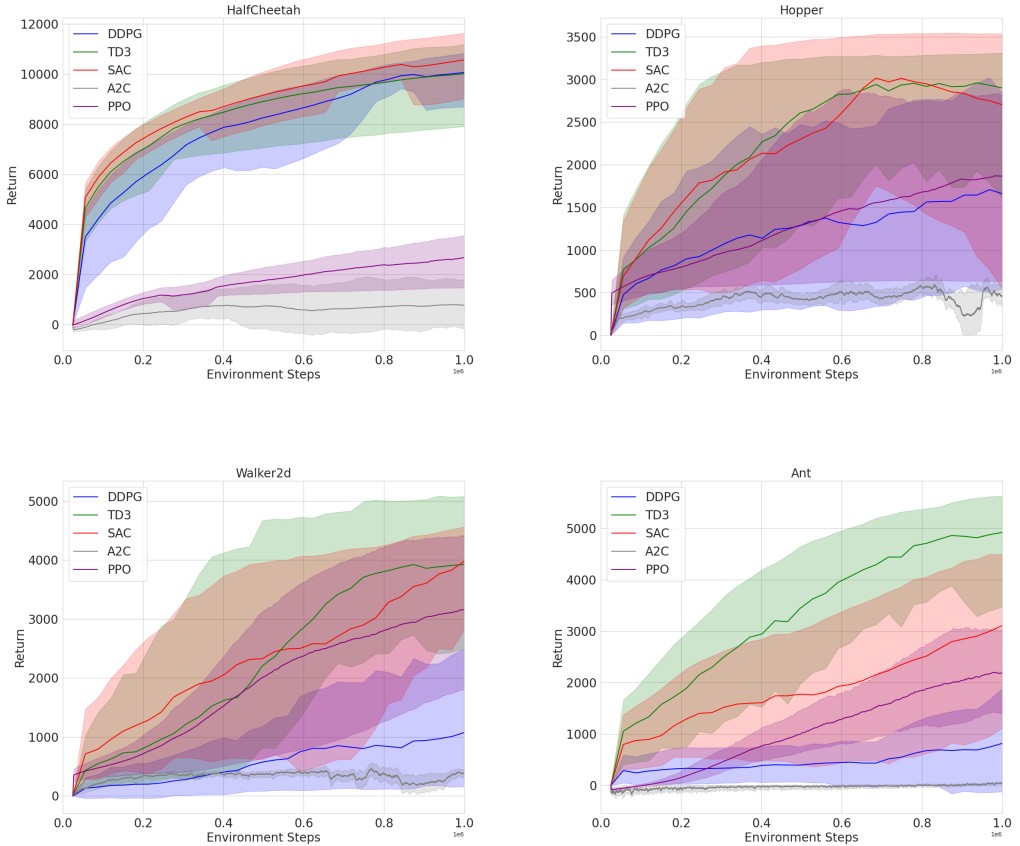

Figure 9: Online RL algorithms trained on MuJoCo environments. We report the mean and standard deviation reward over 5 random seeds. Each was trained for 1M frames.

Table 6: Training parameters for single-agent on-policy algorithms on MuJoCo environments.

| A2C | | PPO | |
|---|---|---|---|
| Discount ($\gamma$) | 0.99 | Discount ($\gamma$) | 0.99 |
| GAE $\lambda$ | 0.95 | GAE $\lambda$ | 0.95 |
| num envs | 1 | num envs | 1 |
| Horizon (T) | 2048 | Horizon (T) | 64 |
| Adam lr | $3e^{-4}$ | Adam lr | $3e^{-4}$ |
| Minibatch size | 64 | Minibatch size | 64 |
| Policy architecture | MLP | Policy architecture | MLP |
| Value net architecture | MLP | Value net architecture | MLP |
| Policy layers | [64, 64] | Policy layers | [64, 64] |
| Value net layers | [64, 64] | Value net layers | [64, 64] |
| Policy activation | Tanh | Policy activation | Tanh |
| Value net activation | Tanh | Value net activation | Tanh |
| Critic coef. | 0.5 | Critic coef. | 0.5 |
| Entropy coef. | 0.0 | Entropy coef. | 0.0 |
| | | Num. epochs | 10 |
| | | Clip $\epsilon$ | 0.2 |

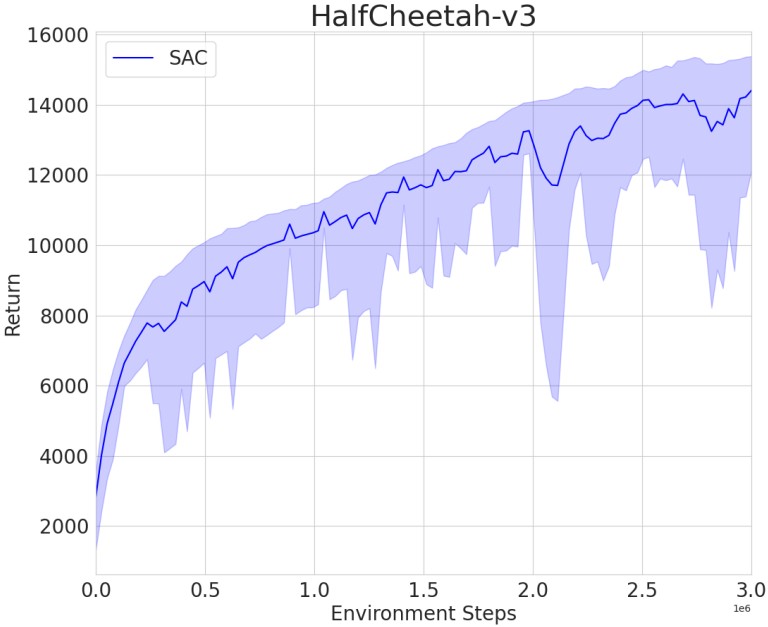

Figure 10: To verify its paper performance we ran SAC on the HalfCheetah-v3 MuJoCo environment for 3M frames. We report the mean and standard deviation reward over 5 random seeds.

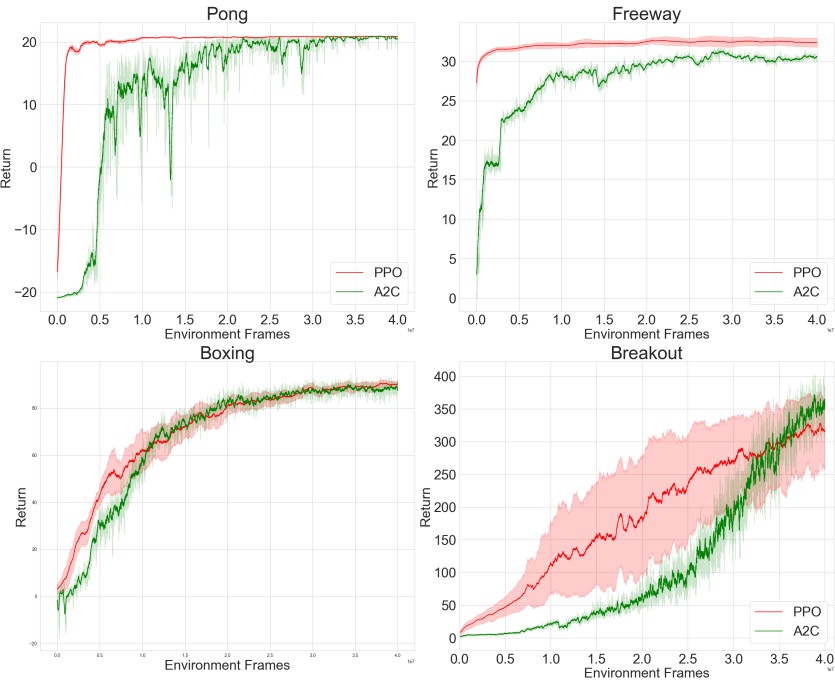

Figure 11: Online RL algorithms with discrete action space trained on Atari environments. We report the mean and standard deviation reward over 5 random seeds. Algorithms were trained for 40M game frames (10 M timesteps since we use frameskip 4).

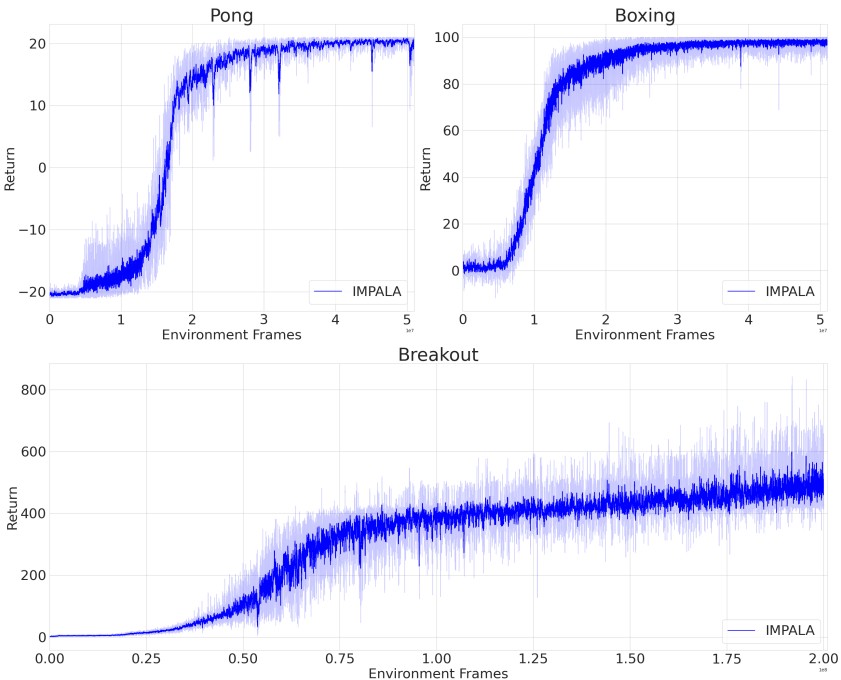

Figure 12: Distributed training results for IMPALA trained on 3 Atari environments. We report the mean and standard deviation reward over 5 random seeds. IMPALA was trained for 200M game frames (50M timesteps since we use frameskip 4) for each environment, however, for the Pong and Boxing environments rewards of only the first 50M frames are visualized as IMPALA converged early. For Freeway, IMPALA is unable to get more that 0.0 reward, both in the original paper and in our experiments.

Table 7: Training parameters for single-agent on-policy algorithms on Atari environments. For all cases, we follow the environment data transforms from DeepMind (Mnih et al., 2013)

| A2C | | PPO | | IMPALA | |
|---|---|---|---|---|---|
| Discount ($\gamma$) | 0.99 | Discount ($\gamma$) | 0.99 | Discount ($\gamma$) | 0.99 |
| GAE $\lambda$ | 0.95 | GAE $\lambda$ | 0.95 | num envs | 1 (12 workers) |
| num envs | 1 | num envs | 1 | Horizon (T) | 80 |
| Horizon (T) | 80 | Horizon (T) | 4096 | RMSProp lr | $6e^{-4}$ |
| Adam lr | $1e^{-4}$ | Adam lr | $2.5e^{-4}$ | RMSProp alpha | 0.99 |
| Minibatch size | 80 | Minibatch size | 1024 | Minibatch size | 32 x 80 |
| Policy architecture | CNN | Policy architecture | CNN | Policy architecture | CNN |
| Value net architecture | CNN | Value net architecture | CNN | Value net architecture | CNN |
| Policy layers | [512] | Policy layers | [512] | Policy layers | [512] |
| Value net layers | [512] | Value net layers | [512] | Value net layers | [512] |
| Policy activation | ReLU | Policy activation | ReLU | Policy activation | ReLU |
| Value net activation | ReLU | Value net activation | ReLU | Value net activation | ReLU |
| Critic coef. | 0.5 | Critic coef. | 0.5 | Critic coef. | 0.5 |
| Entropy coef. | 0.01 | Entropy coef. | 0.01 | Entropy coef. | 0.01 |
| | | Num. epochs | 3 | | |
| | | Clip $\epsilon$ | 0.1 | | |

Table 8: Training parameters for single-agent off-policy algorithms.

| SAC | | TD3 | | DDPG | |
|---|---|---|---|---|---|
| Discount ($\gamma$) | 0.99 | Discount ($\gamma$) | 0.99 | Discount ($\gamma$) | 0.99 |
| Adam lr (all nets) | $3e^{-4}$ | Adam lr (all nets) | $3e^{-4}$ | Adam lr (all nets) | $3e^{-4}$ |
| Batch size | 256 | Batch size | 256 | Batch size | 256 |
| Policy net | MLP | Policy net | MLP | Policy net | MLP |
| Q net | MLP | Q net | MLP | Q net | MLP |
| Policy layers | [256, 256] | Policy layers | [256, 256] | Policy layers | [256, 256] |
| Q net layers | [256, 256] | Q net layers | [256, 256] | Q net layers | [256, 256] |
| Policy activation | ReLU | Policy activation | ReLU | Policy activation | ReLU |
| Q net activation | ReLU | Q net activation | ReLU | Q net activation | ReLU |
| Target polyak | 0.995 | Target polyak | 0.995 | Target polyak | 0.995 |
| Buffer size | 1000000 | Buffer size | 1000000 | Buffer size | 1000000 |
| Init rand. frames | 10000 | Init rand. frames | 10000 | Init rand. frames | 10000 |
| | | Exploration noise | $N(0.0, 0.1)$ | Exploration noise | OU(0.0, 0.2) |
| | | Target noise | $N(0.0, 0.2)$ | $\epsilon$ init | 1.0 |
| | | Noise clip | 0.5 | $\epsilon$ end | 0.1 |
| | | Policy delay | 2 | $\epsilon$ annealing steps | 1000 |
| | | | | $\theta$ | 0.15 |

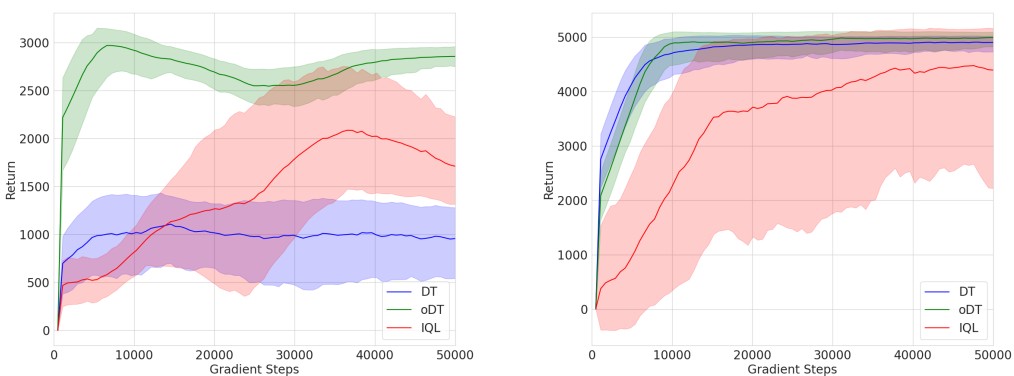

Figure 13: Offline RL algorithms pre-trained on hopper-medium-v2 (left) and halfcheetah-medium-v2 (right) D4RL datasets. We report the mean and standard deviation reward over 3 random seeds. Each run consists of 50,000 network updates.

Table 9: Training parameters for offline algorithms Decision Transformer (DT), Online Decision Transformer (oDT) and Implicit Q-Learning (IQL) for the hopper-medium-v2 (Ho) and halfcheetah-medium-v2 (HC) D4RL datasets.

| DT | | oDT | | IQL | |
|---|---|---|---|---|---|
| Lamb lr | $1e^{-4}$ | Lamb lr | $1e^{-4}$ | Adam lr | $3e^{-4}$ |
| Batch size | 64 | Batch size | 256 | Batch size | 256 |
| Weight decay | $5e^{-4}$ | Weight decay | $5e^{-4}$ | Policy net | MLP |
| Scheduler | LamdaLR | Scheduler | LamdaLR | Q net | MLP |
| Warmup steps | 10000 | Warmup steps | 10000 | Value net | MLP |
| Train context | 20 | Train context | 20 | Policy layers | [256, 256] |
| Eval context Ho | 20 | Eval context Ho | 20 | Q net layers | [256, 256] |
| Eval context HC | 5 | Eval context HC | 5 | Value net layers | [256, 256] |
| Policy net | GPT2 | Policy net | GPT2 | Policy activation | ReLU |
| Embd dim | 128 | Embd dim | 512 | Q net activation | ReLU |
| Hidden layer | 3 | Hidden layer | 4 | Value net activation | ReLU |
| Attn heads | 1 | Attn heads | 4 | Target polyak | 0.995 |
| Inner layer dim | 512 | Inner layer dim | 2048 | temperature ($\beta$) | 3.0 |
| Activation | ReLU | Activation | ReLU | expectile ($\tau$) | 0.7 |
| Resid. pdrop | 0.1 | Resid. pdrop | 0.1 | Discount ($\gamma$) | 0.99 |
| Attn pdrop | 0.1 | Attn pdrop | 0.1 | | |
| Action head | [128] | Action head | [512] | | |
| Reward scaling | 0.001 | Reward scaling | 0.001 | | |
| Target return Ho | 3600 | Target return Ho | 3600 | | |
| Target return HC | 6000 | Target return HC | 6000 | | |
| | | init alpha | 0.1 | | |

### G.2.2 HYPERPARAMETERS

Table 9 displays all hyperparameter values and network architecture details required to reproduce our offline RL results.

## H MULTI-AGENT REINFORCEMENT LEARNING EXPERIMENTS

Most of TorchRL's components are default-compatible with Multi-Agent RL (MARL), and MARL-specific components are also available. The requirement for nested data structures and multi-dimensional batch-size handling make TensorDict the ideal structure to support data representation in MARL contexts. TensorDicts allow to carry both per-agent data (e.g., reward in POMGs (Littman, 1994)) and shared data (e.g., global state, reward in Dec-POMDPs (Bernstein et al., 2002)) by storing the tensors at the relevant nesting level, thus highlighting their semantic difference and optimizing storage. MARL also needs to cope with heterogeneous input/output domains, (ie, settings where the shape of the agent's attributes differ). TorchRL and TensorDict provide appropriate primitives to handle these cases with minimal disruption in the code. As for single-agent solutions, MARL losses can easily be swapped and are semantically similar to their single-agent counterparts.

To showcase TorchRL's MARL capability, we implement six state-of-the-art algorithms and benchmark them on three tasks in the VMAS simulator Bettini et al. (2022). Unlike existing MARL benchmarks (e.g., StarCraft Samvelyan et al. (2019), Google Research Football Kurach et al. (2020)), VMAS provides vectorized on-device simulation and a set of scenarios focused on continuous and partially observable multi-robot cooperation tasks. TorchRL and VMAS both use a PyTorch backend, enabling performance gains when both sampling and training are run on-device. We implement MADDPG Lowe et al. (2017), IPPO de Witt et al. (2020), MAPPO Yu et al. (2022), IQL Tan (1993), VDN Sunehag et al. (2018), and QMIX Rashid et al. (2020). This selection of algorithms presents many of the different flavors discussed in this section. For uniform evaluation, we perform all training on-policy with the same hyperparameters and networks. For algorithms that require discrete actions, we use the default VMAS action discretization (which transforms 2D continuous actions into 5 discrete directions). We evaluate all algorithms in three environments: (i) *Navigation* (Figure 14a),

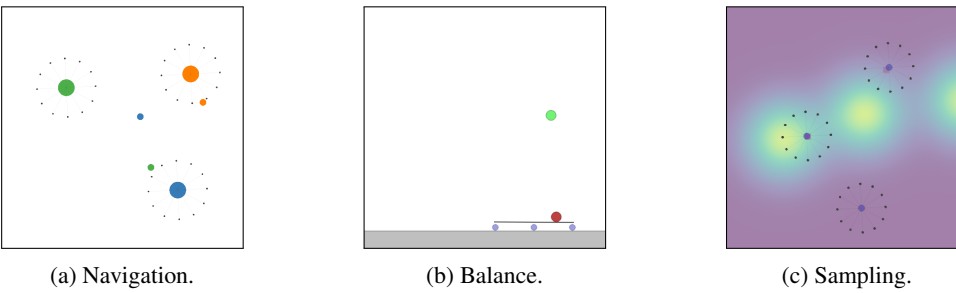

(a) Navigation.        (b) Balance.        (c) Sampling.

Figure 14: The three VMAS multi-robot control tasks used in the experiments.

where agents need to reach their target while avoiding collisions using a LIDAR sensor, (ii) *Balance* (Figure 14b), where agents affected by gravity have to transport a spherical package, positioned randomly on top of a line, to a given goal at the top, and (iii) *Sampling* (Figure 14c), where agents need to cooperatively sample a probability density field. More details on the tasks, hyperparameters, and training scripts are available in the additional material. Figure 2 shows the results. In all tasks, we can observe the effectiveness of PPO-based methods with respect to Q-learning ones, this might be due to the suboptimality of discrete actions in control tasks but also aligns with recent findings in the literature Yu et al. (2022); de Witt et al. (2020). In the *sampling* scenario, which requires more cooperation, we see how IQL fails due to the credit assignment problem, while QMIX and VDN are able to achieve better cooperation.

Overall, our experiments highlight how different MARL solutions can be compared on diverse tasks with minimal user effort. In fact, the scripts used for this comparison present minimal differences (mainly only in the loss and policy class installations), proving TorchRL to be a flexible solution also in MARL settings.

## H.1 COMPARISON WITH RLLIB

To further assess the correctness of our implementations, we compare the IPPO algorithm (de Witt et al., 2020) with the RLlib (Liang et al., 2018) library on the three VMAS tasks in Figure 14. For the RLlib implementation, we use the code provided with (Bettini et al., 2023). The TorchRL code is publicly available in the repository examples. The comparison, shown in Figure 15, demonstrates that, while TorchRL and RLlib achieve similar returns, TorchRL takes significantly lower time thanks to its ability to leverage the vectorization of the VMAS simulator.

## H.2 IMPLEMENTATION DETAILS

We now describe the implementation details for our experiments.

### H.2.1 HYPERPARAMETERS

To uniform the training process across all algorithms, we train all algorithms on-policy. The hyperparameters used are the same for all experiments and are shown in Table 10. In general, the training scripts have the following structure; There is an outer loop performing sampling. At each iteration, *Batch size* frames are collected using *# VMAS vectorized envs* with *Max episode steps*. $\frac{Batch\ size}{Minibatch\ size}$ optimization steps are then preformed for *SDG Iterations* using an Adam optimizer. The training ends after *# training iterations*. Critics and actors (when used) are two-layer MLPs with 256 cells and tanh activation. Parameters are shared in all algorithms apart from MADDPG to follow the original paper implementation.

### H.2.2 ENVIRONMENTS

The tasks considered are scenarios taken from the VMAS Bettini et al. (2022) simulator. They all consider agents in a 2D continuous workspace. To move, agents take continuous 2D actions which represent control forces. Discrete action can be set and will map to the five options: up, down, left,

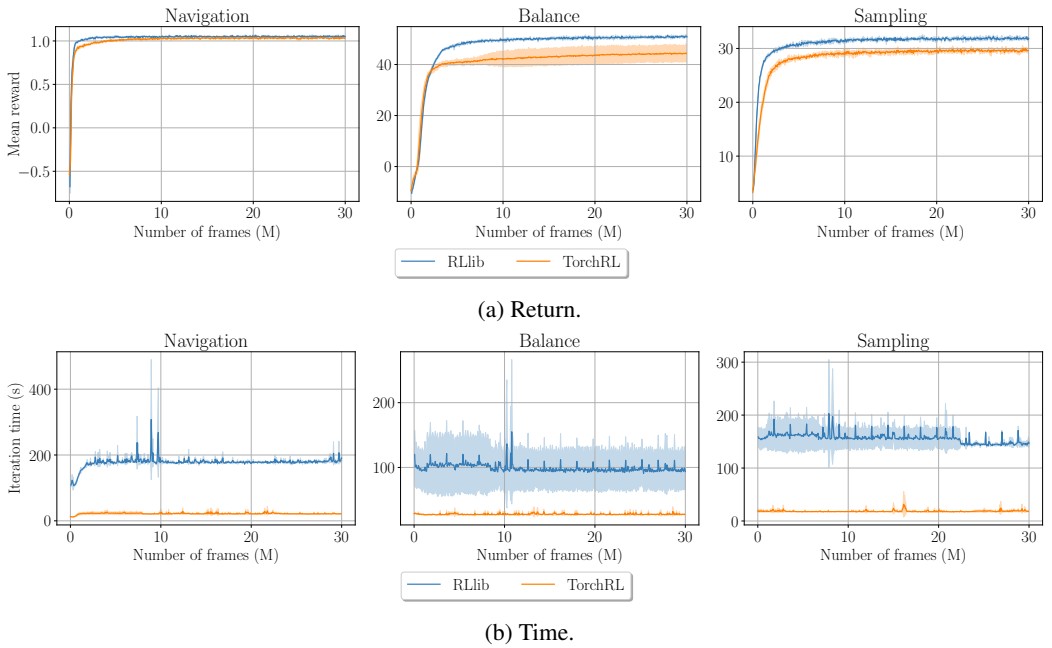

(a) Return.

(b) Time.

Figure 15: Comparison of TorchRL and RLlib on three VMAS tasks using the IPPO algorithm (de Witt et al., 2020). The comparison shows that while TorchRL and RLlib achieve similar returns, TorchRL takes significantly less time thanks to its ability to leverage the vectorization of the VMAS simulator. We report the mean and standard deviation episodic return over 3 random seeds. Each run consists of 500 iterations of 60,000 steps each, with an episode length of 100 steps.

Table 10: MARL training parameters.

| Training | | General | | PPO | |
|---|---|---|---|---|---|
| Batch size | 60000 | Discount $\gamma$ | 0.9 | Clip $\epsilon$ | 0.2 |
| Minibatch size | 4096 | Max episode steps | 100 | GAE $\lambda$ | 0.9 |
| SDG Iterations | 45 | NN Type | MLP | Entropy coeff | 0 |
| # VMAS vectorized envs | 600 | # of layers | 2 | KL coeff | 0 |
| Learning rate | 5e-5 | Layer size | 256 | | |
| Max grad norm | 40 | Activation | Tanh | | |
| # training iterations | 500 | # agents | 3 | | |

right, and staying still. In the following, we describe in detail the three environments used in the experiment:

- *Navigation* (Figure 14a): Randomly spawned agents (circles with surrounding dots) need to navigate to randomly spawned goals (smaller circles). Agents need to use LIDARs (dots around them) to avoid running into each other. For each agent, we compute the difference in the relative distance to its goal over two consecutive timesteps. The mean of these values over all agents composes the shared reward, incentivizing agents to move towards their goals. Each agent observes its position, velocity, lidar readings, and relative position to its goal.

- *Balance* (Figure 14b) Agents (blue circles) are spawned uniformly spaced out under a line upon which lies a spherical package (red circle). The team and the line are spawned at a random $x$ position at the bottom of the environment. The environment has vertical gravity. The relative $x$ position of the package on the line is random. In the top half of the environment, a goal (green circle) is spawned. The agents have to carry the package to the goal. Each agent receives the same reward which is proportional to the distance variation between the package and the goal over two consecutive timesteps. The team receives a negative reward of $-10$ for making the package or the line fall to the floor. The observations for each agent are: its position, velocity, relative position to the package, relative position to the line, relative position between package and goal, package

velocity, line velocity, line angular velocity, and line rotation $\mathrm{mod}\pi$. The environment is done either when the package or the line falls or when the package touches the goal.

- *Sampling* (Figure 14c) Agents are spawned randomly in a workspace with an underlying Gaussian density function composed of three Gaussian modes. Agents need to collect samples by moving in this field. The field is discretized to a grid (with agent-sized cells) and once an agent visits a cell its sample is collected without replacement and given as a reward to the whole team. Agents can use a lidar to sense each other in order to coordinate exploration. Apart from lidar, position, and velocity observations, each agent observes the values of samples in the 3x3 grid around it.

## I  COMPARISON OF DESIGN DECISIONS

As a conclusion note, we provide some more insight into the differences between TorchRL and other libraries UX and design choices.

In contrast with Stable-baselines(Raffin et al., 2021) or EfficientRL (Liu et al., 2021), TorchRL aims to provide researchers with the necessary tools to build the next generation of control algorithms, rather than providing precisely benchmarked algorithms. For this reason, TorchRL's code will usually be more verbose than SB3's because it gives users full control over the implementation details of their algorithm. For example, TorchRL does not make opinionated choices regarding architecture details or data collection setups. Nevertheless, the example repertory and tutorials are available to assist those who wish to get a sense of what configurations are typically deemed more suitable. The trainer class and multiple examples, tutorials, and rich documentation are available to help users get started with their specific problems.

On a similar note, we note that the Tianshou (Weng et al., 2022) entry point for most algorithms is usually a `Policy` class that contains an actor, possibly a value network, an optimizer, and other components. In TorchRL, these items are kept separate to allow users to orchestrate these components at will.

Another difference between TorchRL and other frameworks lies in the data carriers they use: relying on a common class to facilitate inter-object communication is not a new idea in the RL and control ecosystem. Nevertheless, we believe that TensorDict elegantly brings features that will drive its adoption by the ML community. In contrast, Tianshou's `Batch` has a narrower scope than `TensorDict`. Whereas the former is tailored for RL, the latter can be used across ML domains. Additionally, `TensorDict` has a larger set of functionalities and a dedicated `tensordict.nn` package that blends it within the PyTorch ecosystem as shown above.

Next, TorchRL is less an extension of a simulator than other libraries. For instance, Tianshou mainly supports Gym/Gymnasium environments (Brockman et al., 2016) while TorchRL is oblivious to the simulation backend and works indifferently with DeepMind control, OpenAI Gym, or any other simulator.

Finally, TorchRL opts for a minimal set of core dependencies (PyTorch, NumPy, and tensordict) but a maximal coverage of optional external backends whether it is in terms of environments and simulators, distributed tools (Ray or submitit), or loggers. Restricting the core dependencies comes with multiple benefits, both in terms of usability and efficiency: we believe that RLlib's choice of supporting multiple ML frameworks severely constrains code flexibility and increments the amount of code duplication. This has a significant impact on its `SampleBatch` data carrier, which is forced to be a dictionary of NumPy arrays, thus leading to multiple inefficient conversions if both sampling and training are performed on GPU.

## J  ACKNOWLEDGMENTS

We express our gratitude to the numerous contributors who have made TorchRL a reality through their dedicated efforts, whether by directly contributing to the codebase or by providing valuable feedback and suggestions for its development. A complete list of these individuals can be found on the TorchRL GitHub page. Furthermore, we extend our appreciation to the PyTorch and Linux foundations for their support and confidence in this project, as they provide an ecosystem where the library can thrive (e.g., CI/CD, compute resources and technical support).

