# OpenReview forum: "TorchRL: A data-driven decision-making library for PyTorch"
_ICLR.cc/2024/Conference — ICLR 2024 spotlight_

### Official Review · Reviewer_gnLZ · 2023-10-30

**Soundness:** 3 good
**Presentation:** 3 good
**Contribution:** 3 good
**Rating:** 6
**Confidence:** 5

**Summary:**

The paper introduces an new _modular_ reinforcement learning in PyTorch named TorchRL whose goal is provide a flexible, efficient and scalable library for rapid prototyping and research.
The key componnet here the TensorDict data structure enablibg easy efficient communication between different components like environments, buffers, models etc.
It includes many reference implementations including RL algorithms as well as distributed traning and other utilities.
Experiments are performed to validate correctness and efficiency while showing competitive performance compared to other libraries.

**Strengths:**

- Modular design: seems really well designed from the perspective of allowing new implementations from existing components vs the current standard practice of trying to fork an existing implementation where which component changed and mattered can be difficult to figure out. You either have single file implementations that don't really do distributed scaling well or you have weird nested inference to figure out how/where even is the actual algorithm implemented.
- Modularity is demonstrated with reasonable tradeoffs for a very wide variety of RL methods and applications.
- TensorDict as a concept is very generally useful and will hopefully see wider adoption in Torch ecosystem.
- Distributed training and scaling building on `torch.distributed` is quite nice.
- Example implementations are short and clear while scaling well.

**Weaknesses:**

- Likely steeper learning curve and verbosity compared to higher level libraries. However, given the focus on research over applications, should not be that bad.
- Limited to PyTorch, although DLPack has made things easier.
- Smaller community and ecosystem. Would have been much better if this had been ready a couple of years ago that would have helped fix some of the fragmentation issues. If I were to guess, RL _algorithm_ research is somewhat waning (and has largely been not _that_ successful as a research endeavor because a lot of things that make RL work are considered engineering details).
- Lots of tiny details matter when it comes to RL comparisons and more tooling and integration here would be useful. Reporting 3 average seed results is likely not the best way to report results when comparing RL methods.
- Range of "reasonableness" is quite wide when it comes to RL implementations. Would be great if the paper also put the current best reported results for various environments to contextualize. That is also going to be helpful in convincing new algorithm implementations to start here rather than the best performing implementation in terms of reward scores etc.

**Questions:**

- While I understand it's not the focus, would be useful to also show how heteregoenous multi-agent problems can be setup in TorchRL's API while still potentially leveraging batching/vectorization capabilities?
- Hyperparameters in RL are a big bane. Any plans to support automatic tuning, especially population based methods?

---

> ### Author Response · Authors · 2023-11-16
>
> Thank you for your insightful review and valuable questions regarding our paper. We appreciate the opportunity to address your concerns and clarify various aspects of our research.
>
> Regarding the learning curve and verbosity of TorchRL, one of our goals is to be a service provider for other libraries and RL developers who are likely willing to get a deep understanding of the code underpinning their implementations. For this reason, we make sure that we do not only document how to use the library but also how to customize it. We are aware that this introduces some extra information that may not be relevant to all users. In the future, we plan on separating the documentation for basic and advanced users in order to solve that problem.
>
> On the library being limited to PyTorch, our focus is to maintain PyTorch as the core of RL research; hence, we currently do not plan to extend support to other backends like Jax or DLPack.
>
> Regarding the RL community and ecosystem, we acknowledge the perception of RL research waning in recent years. However, we hold an optimistic view. RL continues to be of interest in ML and AI developments, with significant progress in foundational models and offline learning techniques. This progress has translated into various real-world applications, like robotics, drug discovery or gaming, demonstrating RL's evolving maturity. We believe the adoption of a unified and genuinely generalistic decision-making framework like TorchRL has the potential to reignite enthusiasm and momentum in the field of RL research. We are very enthusiastic about the future of RL and related fields and hope that this library will help bring these solutions to life.
>
> With respect to your concerns regarding the nuances in RL comparisons, we have expanded our experiments to include 2 additional seeds. By providing results based on a total of  5 seeds, we aim to offer a more thorough and reliable evaluation. At the moment of writing this response, we are running the experiments and will update the paper by the end of the discussion period. Early results from these new seeds support our initial results. Regarding tooling and integration, our goal is to provide an extensive toolbox to easily and reliably experiment with how changes in a configuration can improve or deteriorate the results of a specific algorithm.
>
> On providing experiments that replicate the most popular results vs the best performing implementations available:  This is a very valid point. We struggled with the decision of whether to target the best outcomes or strictly replicate the most recognized results in the literature (such as the original implementations of each algorithm). For the paper, we opted for the latter as we believed it would be a more convincing approach to demonstrate the correctness of our implementations, given that they are better known and tested. However, we intend to include state-of-the-art examples in the repository alongside the paper implementation, recognizing the potential interest of many users. We hope our library achieves this difficult balance between showcasing what we already know to work and making it possible to code the “next big thing”.
>
> Addressing the question of heterogeneous multi-agent problems, we believe this is also a great point and it has been the focus of intensive development over this summer. The core functionality introduced was the grouping mechanism, which allows MARL users to decide how to group agents’ data in tensordicts. Data belonging to agents within the same group will be stacked, while data from agents in different groups will be kept as separate entries. This way users can decide whether to benefit from vectorization in the agent dimension, by grouping agents together, or whether to keep data separate to allow for heterogeneity. In any case, all groups can be vectorized over an environment dimension, meaning that heterogeneity choices will not impact environment vectorization benefits. While we have already produced examples and tutorials for homogeneous agents (all stacked in the same group), it is in our priorities to produce tutorials and examples for heterogeneous agents. It is also important to notice that this paradigm is already used in the BenchMARL library, which uses the TorchRL MARL grouping API to allow the training of homogeneous and heterogeneous tasks from VMAS and PettingZoo.
>
> Finally, on the topic of hyperparameter tuning in RL, we acknowledge its critical role. While automatic tuning, particularly through population-based methods, is not currently a feature of our library, we recognize its potential and are considering it for future updates. In the meantime, we support robust logging tools like Weights & Biases for manual hyperparameter optimization.
>
> We hope that these responses address your concerns and questions.

---

> > ### Author Response · Authors · 2023-11-20
> >
> > Expanding on our initial response, the paper has now been updated to include 5-seed results in the tables and plots that previously only included 3.

---

### Official Review · Reviewer_tnNy · 2023-10-30

**Soundness:** 2 fair
**Presentation:** 2 fair
**Contribution:** 3 good
**Rating:** 6
**Confidence:** 3

**Summary:**

This paper introduces TorchRL, a PyTorch-based reinforcement learning/sequential decision-making library. The library supports many standard algorithms in model-free deep RL including the DQN family of algorithms, SAC, and PPO. Additionally, the library offers several additional features/algorithms that have emerged only in the past 2-3 years, including RLHF and decision transformers. There is also additional support for offline RL algorithms, multi-agent algorithms, and model-based RL algorithms such as Dreamer.

Core to TorchRL is another contribution of the paper: the TensorDict (released as a second library), which is a flexible data carrier for PyTorch. It is a dictionary-style object that stores tensor-like objects.

The library is well-tested with good adoption by the community. Several standard algorithms from TorchRL are benchmarked in terms of performance and speed, demonstrating that TorchRL runs much faster than other RL libraries.

**Strengths:**

- Comprehensivity: The library is extremely comprehensive. It spans different algorithm classes (e.g., offline RL, model-free Deep RL, multi-agent RL, etc.) whereas most algorithms focus on a single algorithm class (typically model-free RL). This makes this library more likely to be a single place where researchers and practitioners can go for a solution to their problem.
- Modernity: This library includes multiple powerful tools introduced only in the past 2-3 years, including RLHF and decision transformers.

**Weaknesses:**

Presentation: I did not find the explanation of TensorDict, which seems integral to the paper to be very didactic. The text delves into the various functionalities of the TensorDict at a high-level, but I did not find it clear how this abstraction or these functionalities lends itself to ease development of RL algorithms. Figure 1 did not clarify this. I realize there is more context provided in the appendix, but the explanation in the main text needs to be improved.

Benchmarking: For a software library, benchmarking is extremely important. It is well known that implementation details are critical for performance, making high quality algorithm implementations a core aim of deep RL libraries. All the experiments are run with a mere 3 seeds. Moreover, the Atari results are run on 40M timesteps as opposed to the usual 50M. Moreover, for some environments/algorithms, the performance seems slightly worse than these should be. For example, if we compare the SAC results from Table 1 to PFRL (https://github.com/pfnet/pfrl/tree/master/examples/mujoco/reproduction/soft_actor_critic), we see:
- HalfCheetah: TorchRL performs at 11419 vs. around 15000 in PFRL and other papers
- Ant: TorchRL scores about 3692 as opposed to the 5800+ seen in PFRL and other papers.

SAC also slightly underperforms in the other two environments though the differences are not profound. Is there an explanation for this and for other environments? How are the hyperparameters found for these implementations? How do these implementations differ from other libraries?

Of course, the results I am linking appear to be from the v2 of the environments, whereas the results here in the paper are from v3, which may explain the results. Do you know or have a citation for good results to be expected on the v3 mujoco environments? It is also possible that these differences are due to randomness, which is all the more reason to run more random seeds for benchmarking.

**Questions:**

- The introduction has strong statements indicating that other libraries are not being actively maintained or having fallen out of favor. Active maintenance is easier to see through version control, but how have the authors concluded that other libraries have fallen out of favor? My understanding is that many of these libraries are still being used.
- For replay buffers, the paper says: "To overcome this, TorchRL provides a single RB implementation that offers complete composability. Users can define distinct components for data storage, batch generation, pre-storage and post-sampling transforms, allowing independent customization of each aspect." Is there support for prioritized experience replay which is used in several model-free RL algorithms?

---

> ### Author Response · Authors · 2023-11-16
>
> Thank you for your feedback and insightful questions regarding our paper. We appreciate the opportunity to clarify and enhance our work in response to your review.
>
> Regarding the presentation of TensorDict, we understand the need for a clearer introduction and contextualization of TensorDict in the main text. To address this, we've revised the explanation, starting by introducing the challenges in coding decision-making algorithms and then transitioning to explain how TensorDict addresses these issues. This approach should provide the reader with a better context before diving into TensorDict's features.
>
> On the benchmarking front, we recognize the critical importance of robust benchmarking for software libraries. As per your suggestion, we have expanded our experiment runs to include a total of 5 seeds for each experiment. This should provide a more reliable assessment of performances and enhance the robustness of our results. At the moment of preparing this response, we are in the process of running the experiments and plan to update the paper by the end of the discussion period. Initial results from these new seeds align with our original findings.
>
> Concerning the Atari results, our choice of 40 million steps aligns with the methodology in the "Proximal Policy Optimization Algorithms" paper by Schulman et al., which we used as a reference point to replicate results for A2C and PPO and ensure the correctness of our implementations. This approach is consistent with some studies in the field, although we acknowledge that 50 million steps are also common. We want to clarify that the choice is not arbitrary.
>
> In response to the performance comparisons with other environments and algorithms, we have run additional tests and can confirm that our implementations and hyperparameter choices are in line with established practices. We looked specifically at the SAC results mentioned for HalfCheetah in PFRL. Here the final performance values are for 3 million environment steps. However, we limited our simulations to 1 million steps similar to the TD3 paper, to optimize time due to constraints. This is also consistent with the main convergence pattern of off-policy algorithms in MuJoCo environments. When comparing the plots of PFRL and TorchRL for HalfCheetah, up to 1 million steps they lay in the same range of performance. Further, as mentioned, our results are obtained using the v3 of MuJoCo environments. We chose this version to match the version of environments at the time of the paper release. To directly compare the results of our implementation we additionally ran the HalfCheetah-v2 for 5 seeds and 1M steps. The results of v3 and v2 for HalfCheetah for 1 million environment steps do not show a significant difference in performance.  As asked we can point to other resources of algorithm benchmarks trained on the MuJoCo v3 like Open AI spinning up https://spinningup.openai.com/en/latest/spinningup/bench.html. Important to note, that developers used slightly different hyperparameter settings, like a higher learning rate for example.
>
> Regarding the maintenance and popularity of other libraries, we have revised our statements to focus solely on the aspect of active maintenance, removing any assertions about their popularity or favorability among users.
>
> Lastly, for replay buffers, TorchRL not only supports prioritized experience replay but our implementation is backed by an efficient C++ implementation of Sum and MinTree that allows for a fast, multithreaded retrieval of priorities for millions of items. The option to use prioritized experience replay or not can be directly selected in the configuration of each off-policy algorithm. Regarding the phrase "single RB implementation," we intend to convey that TorchRL utilizes a unified Replay Buffer class, which can be configured with a variety of storage options (such as physical memory and lists), sampling methods (including circular and prioritized sampling), along with different writers and transformation capabilities. This flexible architecture allows us to offer numerous Replay Buffer implementations.
>
> We hope these clarifications and enhancements address the points you've raised, and we look forward to further discussions on our work.

---

> > ### Comment · Reviewer_tnNy · 2023-11-19
> > **Response to Author comments**
> >
> > Thank you for your detailed response.
> >
> > > Regarding the presentation of TensorDict, we understand the need for a clearer introduction and contextualization of TensorDict in the main text. To address this, we've revised the explanation, starting by introducing the challenges in coding decision-making algorithms and then transitioning to explain how TensorDict addresses these issues. This approach should provide the reader with a better context before diving into TensorDict's features.
> >
> > Is this reflected in a revision? (just before I expend time searching for it)
> >
> > > Concerning the Atari results, our choice of 40 million steps aligns with the methodology in the "Proximal Policy Optimization Algorithms" paper by Schulman et al., which we used as a reference point to replicate results for A2C and PPO and ensure the correctness of our implementations. This approach is consistent with some studies in the field, although we acknowledge that 50 million steps are also common. We want to clarify that the choice is not arbitrary.
> >
> > I understand, but I think you misinterpret the results from the PPO paper. See the caption in Table 6 of the PPO paper (https://arxiv.org/pdf/1707.06347.pdf). They clearly state 10M timesteps and 40M frames. Here in your response you are stating 40M timesteps, which is neither what is done in DQN-style papers nor the PPO paper.
> >
> > When I look at Table 1 of the paper (i.e., the TorchRL paper), it says 40M "timesteps". Figure 10 is a bit contradictory and says the algorithms are trained for 40M "frames", but then the actual plots very clearly have on the x-axis "Environment steps" for 40M. Something is incorrect here and should be fixed. It does make me concerned about the benchmarking. Are the authors aware of frameskips? Can you clarify what's going on here?
> >
> >
> > > Here the final performance values are for 3 million environment steps. However, we limited our simulations to 1 million steps similar to the TD3 paper, to optimize time due to constraints.
> >
> > This does clarify things, and I suppose it helps make the figure more readable. I would definitely urge the authors to confirm the performance of SAC against the training length used in actual SAC paper, as opposed to the TD3 paper, for reproducibility purposes. Even if the results are in the appendix.
> >
> > > The results of v3 and v2 for HalfCheetah for 1 million environment steps do not show a significant difference in performance.
> > Thanks for checking this.
> >
> > For the items I have not responded to, that is because I found the author response to ameliorate my concerns. So thank you for that.

---

> ### Author Response · Authors · 2023-11-20
>
> Thank you very much for your follow-up comments.
>
> In response to whether the TensorDict has been updated in the paper: Yes, we edited the first paragraph of the TensorDict section to better define the scope of this abstraction in response to your comment. The differences between the new and old versions should be accessible on the openreview console through Revisions -> Compare Revisions -> Show Differences. However, we copy here the initial TensorDict paragraph that has been modified. Now it says:
>
> """
> Introducing a seamless mode of communication among independent RL algorithmic components poses a set of challenges, particularly when considering the diversity of method signatures, inputs and outputs. A shared, versatile communication standard is needed in such scenarios to ensure fluid interactions between modules without imposing constraints on the development of individual components. We address this problem by adopting a new data carrier for PyTorch named TensorDict, packaged as a separate open-source lightweight library. This library enables every component to be developed independently of the requirements of the classes it potentially communicates with.
> """
>
> Regarding the results from the PPO paper: We now understand better that the confusion comes from our explanation, in which we used “timesteps” and “frames” without a clear-cut distinction. For the results in our paper,  we run the experiments for 10M timesteps with frameskip 4, as in the original PPO paper, and therefore for a total of 40M frames. We then plot the results along a 40M X axis representing the frames as it is done also in the PPO paper. We have now fixed table 1 text so it specifies 40M “frames”. Figure 10 also now specifies that we trained for  “40M frames (10 M timesteps since we use frameskip 4)”, and the X axis has been changed to “environment frames”. We have similarly fixed figure 11, in which we explain that IMPALA is trained for “200M frames (50 M timesteps since we use frameskip 4)”. Thank you for helping us realize the error in our explanation.
>
> Finally, regarding the performance of SAC: We are conducting a final test for SAC with 3M steps on the HalfCheetah environment, which will be completed tomorrow. We plan to make a final update of the PDF to include these results in the appendix.
>
> Besides that, the paper has already been updated to include 5-seed results in the tables and plots that previously only included 3.

---

> > ### Author Response · Authors · 2023-11-21
> >
> > To complete our previous response, we have now added in the appendix SAC results for HalfCheetah-v3 to verify the results are aligned with the original SAC paper (5 seeds).
> >
> > Note that the new figure becomes figure 10, and thus the PPO/A2C results for Atari become figure 11, and the IMPALA results figure 12.

---

> > > ### Comment · Reviewer_tnNy · 2023-11-22
> > > **Changed Score**
> > >
> > > Thanks for addressing these issues. I have adjusted my score to a 6. Honestly I can't say I fully grasped the utility (and more importantly how it's critical) of the TensorDict from the updated explanation, but I do not believe it's a significant enough of a problem (in light of addressing all of the other concerns) to warrant rejection given all the other positive aspects of the paper.
> > >
> > >  I will try to reread that portion of the paper again.

---

### Official Review · Reviewer_77b1 · 2023-11-01

**Soundness:** 4 excellent
**Presentation:** 3 good
**Contribution:** 4 excellent
**Rating:** 8
**Confidence:** 4

**Summary:**

The paper introduces TorchRL, a comprehensive decision-making library in the PyTorch ecosystem, designed to address fragmentization in the decision-making field by providing efficient, scalable, and flexible primitives suitable for complex real-world data and environments. The proposed library offers a modular design, combining components that maintain standalone functionality.  TorchRL introduces several core modular classes as well as TensorDict, a standalone library that can handle multiple tensors at the same time in the form of an improved dictionary class that supports batched tensor operations. An extensive experimental section demonstrates the correct reproduction of several famous single/multi-agent and offline RL as well as the library’s efficiency against strong competitors.

**Strengths:**

The proposed library tries to tackle an important problem - the fragmentization of the RL field. While many libraries have been proposed over the years (in the form of the environment “Gyms” or full-stack RL solutions), TorchRL is, to the best of my knowledge, the most comprehensive and efficient with full support in the PyTorch ecosystem, which makes it exceptionally relevant given the prevalence of PyTorch implementations in the research community, as well as for practitioners. The experimental section touches on virtually all of the most important aspects for benchmarking, single-agent, multi-agent, and offline RL, as well as other benchmarks. The Github library has also received widespread support from several groups.

Finally, I have personally used TorchRL and TensorDict in several projects and found them useful and efficient. I expect the library to have a great impact in streamlining the development of decision-making algorithms for practitioners and researchers alike.

**Weaknesses:**

The paper and its proposed library do not have any major weaknesses.

The main weakness - that does not influence my score since it is highly subjective, due to my experience while using it - is that the library’s implementation is at times hard to read - meaning that the code is at times a bit convoluted and not straightforward to understand. For instance, other platforms such as CleanRL are way easier, in my opinion (also due to their single-file implementations, which are orthogonal to the proposed library), in terms of how the whole RL process works from data collection to training, and as such perhaps more suitable for new researchers.

---

Typos (no influence to my score)

- Page 4, `, Isaac makoviychuk2021isaac, Jumanji (Bonnet et al.),` missing `\cite` and possibly the year for Jumanji
- Page 22 . `implementations1`

**Questions:**

1. Tables 3-5 show a performance comparison on a machine with 8 GPUs. Do the tested gaps with other libraries hold in single-GPU machines as well? In other words, could the gaps be in part due to the fact that TorchRL is using multiple GPUs?
2. In the Appendix, Figure 14, it looks like TorchRL can outperform RLLib in speed but cannot reach the same rewards. Does it mean there may be some incorrect implementation?
3. In Table 4, it is mentioned: "Those data moves are unaccounted for in this table but can double the GAE runtime" . Does this apply only to TorchRL?
4. Could you quantify the overhead of using TensorDict a bit more? It would be useful to see (e.g., in Appendix E) how much the difference is between using TensorDict and not using it.
5. You mentioned that some libraries make environments vectorizable and runnable directly on GPUs (i.e., with PyTorch or JAX). In my experience, CPU-GPU overhead is one of the major challenges in terms of efficiency. Are you planning to develop a “Gym-like” library of vectorized environments as well in the future, such as “pendulum” but with `_step` and `_reset` with tensors?

---

> ### Author Response · Authors · 2023-11-16
>
> We appreciate your insightful feedback on our paper and would like to address the concerns and questions you raised.
>
> Firstly, regarding the readability of our library's implementation, we acknowledge that it can be challenging at times. We are committed to enhancing the code's clarity and documentation and are completely open to suggestions from the community, but striking a balance between a modular, flexible design and simplicity is a delicate task. We would like to emphasize that:
> (1) we do not seek the same "single file, all-inclusive" implementation as CleanRL, and instead focus on flexibility, modularity, breadth and reusability.
> (2) Some users find a "trainer" class with a single entry point more readable than self-contained scripts. Others prefer single scripts. A third category likes scripts with a separate “utils” script that contains the constructors and some extra features. It is very hard (if not impossible) to satisfy them all with a single solution.
> In any case, we will make sure to focus more on the readability of our examples in the future and improve the library's documentation at the same time.
>
> Concerning the performance comparison on a machine with 8 GPU in tables 3-5,  This should have been more clearly stated in the manuscript. The node we were using has 8 GPUs, but we only required 1 GPU through the scheduler to run these experiments. We specified the full node configuration for reproducibility purposes on a similar machine on a public cluster. The speed with 8 GPUs would have been better than the one displayed, but we did not show it to stay within reasonable bounds for an average (academic) user without access to a dedicated, recent cluster. We clarify now in the manuscript that we use only 1 GPU.
>
> In response to your question about Figure 14 in the Appendix: in the comparison reported in appendix H.1 we tried to match the exact hyperparameters used in TorchRL and RLlib libraries. However, this is possible only to a certain degree, as RLlib’s implementations do not provide full transparency and an in-depth knowledge of the codebase is needed for a better understanding. The cause for the slight reward mismatch could be due to this, as we are using a very simple training loop for TorchRL while RLlib could be performing some minor extra optimizations.  Additionally, we believe we can exclude an incorrect implementation for two reasons: (1) the PPO implementation used in this experiment is the same used throughout the other experiments in the paper, which shows that its performance matches the expected one on other simulators (2) the difference in rewards could be argued to be negligible (both implementations reach the “solved'' state for the task) and the sample efficiency profiles follow the same shape.
>
> Regarding the question of whether or not the statement in table 4 "Those data moves are unaccounted for in this table but can double the GAE runtime" applies only to TorchRL, the answer is no, we refer to the other libraries. All the libraries we compare against work directly with numpy (ie, the replay buffer stores numpy arrays, gae computation in numpy) and have to cast to pytorch once the batch is ready. Casting the vectors to pytorch takes extra time that is not accounted for in the table. Because TorchRL works directly with pytorch-backed data storage, the casting operation is not needed. However, we decided not to include the casting time in the table because it is not strictly part of the operation.
>
> In response to your inquiry about the overhead of using TensorDict, we have now quantified this and added it to Appendix B for a more comprehensive understanding.
>
> Your point about  CPU-GPU overhead being one of the major challenges in terms of efficiency is a valid one in our experience too, device casting operations are usually responsible for a large chunk of the total overhead. Regarding the environment API, we emphasize that it is aimed at being a possible basis to build environments from the ground up (without relying on Gym), possibly with all operations operated on GPU. We are not currently working on reimplementing specific CPU-based Gym environments to be GPU-based, but would definitely be possible to use them in TorchRL. However, notice that `torch.compile` now allows running numpy-based gym envs directly on GPU, which we expect will facilitate their integration within PyTorch code bases, as well as batched execution of gym environments (through torch.vmap) and execution on gpu. This is an active area of development for us as well as pytorch developers.
>
> Finally, thank you for pointing out the typos; we have rectified them in the revised version of the paper.

---

### Official Review · Reviewer_C5vY · 2023-11-01

**Soundness:** 3 good
**Presentation:** 3 good
**Contribution:** 3 good
**Rating:** 6
**Confidence:** 3

**Summary:**

Authors develop a new unified library for RL training that introduces the TensorDict, which enables clean modular and composable implementations of all major RL algorithms and support for vectorized GPU environments such as IsaacGym and Brax. The TensorDict enables indexing by key like a regular python dict, but also indexing by shapes like a regular pytorch tensor, which allows it to inherit tensor-based operations like reshaping, concatenating, moving between devices, gradient storing, etc. TorchRL also includes TensorDictSequential as an nn.Sequential analogue operating on TensorDicts.

TorchRL has implementations of nn.Modules including MLP, LSTMs, but also RL specific things like ActorCriticOperator.

**Strengths:**

- Writing contains detailed coverage of major existing RL frameworks and environments.
- Substantial library implementation and development that is already having a large impact on the research community.
- TorchRL includes a communication class.
- Proposed TorchRL framework integrates well with existing pytorch code and all facets of RL are considered, including replay buffers, vectorized environments, and wall clock time.

**Weaknesses:**

### Overall
(A1) Lacks a discussion of limitations. What are the current features that torchRL cannot handle/is bad at handling but should be improved upon in future releases?

(A2) Details of TorchRL’s communication class could be more clear. What defines a communication class? What part of downstream RL algorithms does it accelerate?

(A3) Prior work discussion should describe data structures similar to TensorDict that have been implemented in the past on accelerated ML libraries, such as perhaps pytrees in jax, which seem to share some similarities with TensorDict.

### Computational efficiency results are a bit lacking.

(B1) Paper did not measure the computational overhead/efficiency gain of using TensorDicts vs. regular python data structures containing pytorch objects. For instance, what is the efficiency of standard reshaping operations on TensorDicts of some set of tensors $X$, vs. the same reshaping operation on a regular python dict containing the same set of tensors $X$?

(B2) How does an optimized TorchRL implementation compare efficiency-wise to the popularly-used pytorch implementations of different RL algorithms that are publicly available? Table 4-5 mostly compares with other RL frameworks that do not use pytorch. Without these results, it is hard to determine true efficiency effects of TensorDicts.

(B3) A more in-depth computational efficiency comparison between TorchRL and Jax, on the same hardware, could help practitioners decide which library to use (although ease of use is a separate issue that cannot be easily quantifiable). There is a comparison to CleanRL (Jax) in Table 4, but not in Table 5.

**Questions:**

1. Will tensor-based operations only work on TensorDicts if the leading dimensions of every single tensor in the TensorDict shares the same leading dimension? Can these operations only be done on a subset of the items in the TensorDict?
2. What are the next major features of TorchRL under development?
3. How much resources will be dedicated to supporting TorchRL (github issues, new features) for the foreseeable future, compared to PyTorch as a whole?
4. Are there computational efficiency results of TorchRL on accelerator-based simulators such as IsaacGym, Brax, and MJX?
5. Typo in 3rd paragraph under Section 2, Subsection “Environment API, wrappers, and transformations” with the “Isaac” citation.
6. In Section 2, subsection “Environment API, wrappers, and transformations,” how does the data transform involving “embedding through foundation models” “come at no cost in terms of functionality?” What does this sentence mean?

---

> ### Author Response · Authors · 2023-11-16
>
> We agree more emphasis should be put on the library’s current limitations. Some trade-offs are due to design decisions, i.e. TorchRL is a Pytorch-first library, so JAX users would need to do conversions to use it. Modularity and generality also mean that the focus is not just on simplicity but we need to support complex production scenarios. FSDP is one case that we are currently improving (expected for Q1 2024). Better support for torch.compile is also needed and should appear in the same timeframe. We are also working on eliminating any overhead due to TensorDict conversions.  We are finally still actively improving the documentation.
>
> Regarding the details of TorchRL’s communication class:  With “communication class” we are referring to the TensorDict data carrier itself. We acknowledge the poor choice of words. This observation also raises a valid point: the specific role in accelerating downstream RL algorithms could be introduced more explicitly, explaining how TensorDict features help coding RL algorithms. This suggestion aligns with other reviewers' feedback, it is now addressed in the manuscript.
>
> Following the reviewer’s advice, we added in the appendix a more extensive comparison of TensorDict with other solutions across the RL landscape (incl. pytrees).
>
> Additionally, we have also added a detailed benchmark of TensorDict overhead against PyTrees and other solutions in the appendix. We are actively working on resolving existing issues and we have a clear roadmap to address these, which is detailed in our repository.
>
> Regarding efficiency comparisons with other pytorch-based libs, in table 4-5 we compared TorchRL throughput with SB3, Tianshou and RLLib, which all use PyTorch but not TensorDict.
>
> CleanRL not in table 5: The data collection speed measures in table 5 were executed on Gym environments where the code execution was either numpy-based or another backend (eg, ALE). In these scenarios, what we measured was independent of the ML backend (Jax or PyTorch) but related to how data was collected across environments. We would not expect a different throughput with CleanRL, which uses the batched environment API from gym itself.
>
> Tensor-based operations: In general, a TensorDict with an empty batch-size can contain tensors of any shape, and shape-related ops are not available (eg, split, view). If shape-related operations are required over a subset of tensors, there are currently two main solutions:
> 1. Organise tensors semantically: a root tensordict with an empty shape and a nested tensordict with a non-empty shape that accepts shape-related operations. This is how we approach the problem in the MARL API.
> 2. Isolate a set of tensors with `select`, reset the shape, and call a shape-related method. Depending on the use case there may be simpler ways of achieving the same result.
>
> Next major features under development: we discuss the features of the next release openly with the OSS community via social networks and on GitHub. Here is a summary of the upcoming features:
> Better compatibility with other modules (eg, FSDP, PyG) to facilitate integration in RLHF and large models at scale;
> Dataset hub for offline RL (akin to torchvision’s datasets);
> Better integration of LLMs: build RL-based agents, + better integration in the fields of chemistry or gaming.
> Better exportability of our models for sim2real applications.
>
> Dedicated resources: for anonymity, we cannot disclose much info about where the library’s support comes from. Keeping that in mind, we can state that maintenance won’t be an issue for the foreseeable future. We have strong support from the public and private sectors which totalize approx. 2 full-time engineers (+ OpenSource collaborators which totalize 125 people).
>
> Results on accelerator-based simulators: The results in Table 3 are obtained with VMAS, an accelerator-based simulator. It is shown how TorchRL compares to RLLib in leveraging vectorization to obtain higher throughput. Other pytorch accelerator-based simulators will behave similarly (e.g., Isaac or habitat). We did not include these metrics in the paper for the sake of conciseness (ie, these simulators will perform at their best when executed within the library). Jax-based simulators are compatible with TorchRL too, but they will suffer from a consequent jax-to-torch io overhead.Our tutorials provide a simple example of how to code an accelerator-based environment using the TorchRL API.
>
> Regarding the confusing sentence in Section 2, we have now corrected the phrasing.
> What we meant is that using lists of transforms rather than wrappers does not lower the amount of things one can do. A common worry with lists of transforms is that a transform that is not a wrapper cannot access the environment information for instance. TorchRL's transforms have a bidirectional registration mechanism that allows them to access properties of the parent env, thereby bringing flexibility without the cost.
>
> Finally, We have fixed the typo.

---

### Meta-Review · Area_Chair_K3th · 2023-12-19

**Metareview:**

The paper presents TorchRL, a reinforcement learning library in PyTorch, and introduces a new TensorDict primitive. Reviewers appreciated its broad coverage of RL algorithms and effective integration with PyTorch and its potential for dealing with complex environments. However, concerns were raised about insufficient benchmarks and unclear presentation of key concepts (TensorDict in particular). Given these factors, I recommend acceptance.

**Justification For Why Not Higher Score:**

Presentation could be more clear.

**Justification For Why Not Lower Score:**

Useful tool that community should be aware of.

---

### Decision · Program_Chairs · 2024-01-16

Accept (spotlight)